# How to Learn and Generalize From Three Minutes of Data: Physics-Constrained and Uncertainty-Aware Neural Stochastic Differential Equations

**Franck Djeumou**[*]  **Cyrus Neary**[*]  **Ufuk Topcu**
Center for Autonomy, The University of Texas at Austin
{fdjeumou, cneary, utopcu}@utexas.edu

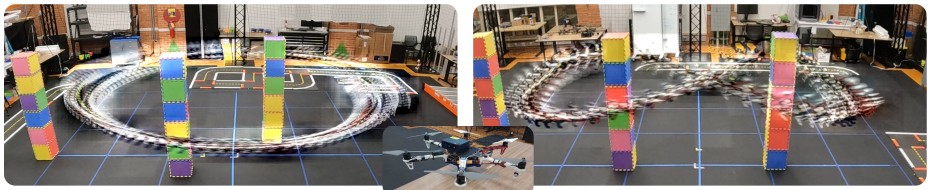

Figure 1: With only three minutes of manually collected data, the proposed algorithms for training neural stochastic differential equations yield accurate models of hexacopter dynamics. When used for model-based control, they result in policies that track aggressive trajectories that push the hexacopter's velocity and Euler angles to nearly double their maximum values in the training dataset. Videos of experiments are available at https://tinyurl.com/29xr5vya.

**Abstract:** We present a framework and algorithms to learn controlled dynamics models using neural stochastic differential equations (SDEs)—SDEs whose drift and diffusion terms are both parametrized by neural networks. We construct the drift term to leverage a priori physics knowledge as inductive bias, and we design the diffusion term to represent a *distance-aware* estimate of the uncertainty in the learned model's predictions—it matches the system's underlying stochasticity when evaluated on states *near* those from the training dataset, and it predicts highly stochastic dynamics when evaluated on states *beyond* the training regime. The proposed neural SDEs can be evaluated quickly enough for use in model predictive control algorithms, or they can be used as simulators for model-based reinforcement learning. Furthermore, they make accurate predictions over long time horizons, even when trained on small datasets that cover limited regions of the state space. We demonstrate these capabilities through experiments on simulated robotic systems, as well as by using them to model and control a hexacopter's flight dynamics: A neural SDE trained using only three minutes of manually collected flight data results in a model-based control policy that accurately tracks aggressive trajectories that push the hexacopter's velocity and Euler angles to nearly double the maximum values observed in the training dataset.

**Keywords:** Neural SDE, Physics-Informed Learning, Data-Driven Modeling, Dynamical Systems, Control, Model-Based Reinforcement Learning

## 1 Introduction

Leveraging physics-based knowledge in learning algorithms for dynamical systems can greatly improve the data efficiency and generalization capabilities of the resulting models, even if the dynamics are largely unknown [1]. However, before such physics-constrained learning algorithms may be used for model-based reinforcement learning (RL) or control, methods to estimate the uncertainty in the model's predictions are required [2, 3, 4, 5].

---

[*]These authors contributed equally.

7th Conference on Robot Learning (CoRL 2023), Atlanta, USA.

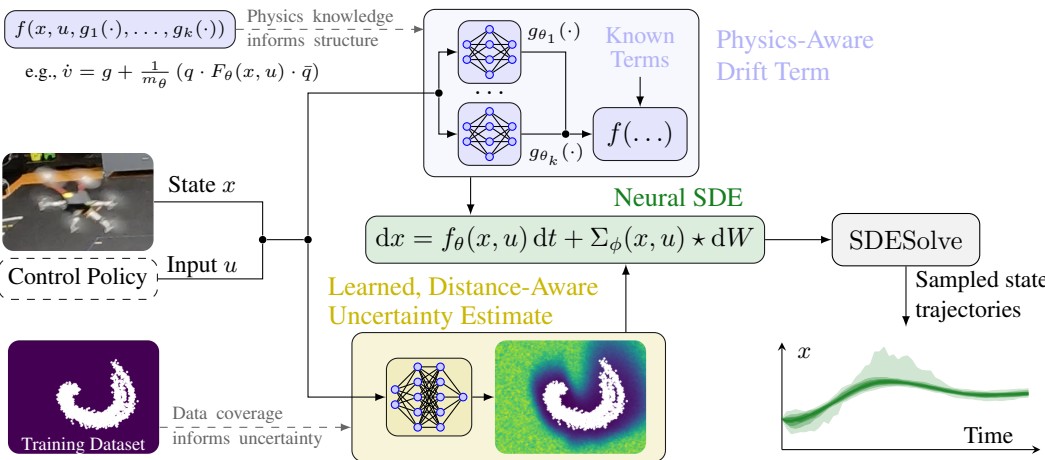

Figure 2: Overview. We propose neural stochastic differential equations (SDEs) to predict system dynamics. The SDE's drift and diffusion terms are both parametrized by neural networks. The drift term leverages a priori knowledge into the structure of the corresponding neural network. The diffusion term is trained to approximately capture the uncertainty in the model's predictions.

To build uncertainty-aware, data-driven models that also leverage a priori physics knowledge, we present a framework and algorithms to learn controlled dynamics models using *neural stochastic differential equations (SDEs)*—SDEs whose drift and diffusion terms are both parametrized by neural networks. Given an initial state and a control signal as inputs, the model outputs sampled trajectories of future states by numerically solving the SDE. Figure 2 illustrates an overview of the proposed approach, which comprises the following two major components.

*Physics-constrained drift term.* We represent the drift term as a differentiable composition of known terms that capture a priori system knowledge (e.g., the rigid body dynamics for the hexacopter from Figure 1), and unknown terms that we parametrize using neural networks (e.g., the nonlinear motor-to-thrust relationship, and unmodeled aerodynamic effects). Even if the system's dynamics are largely unknown, this approach imposes useful inductive biases on the model: Unknown terms are parametrized separately and their interactions are defined explicitly.

*Uncertainty-aware diffusion term.* Meanwhile, we construct the diffusion term to provide a *distance-aware* estimate of the uncertainty in the learned model's predictions—the neural SDE's predictions should have low stochasticity when it is evaluated on states similar to those from the training dataset, and they should be highly stochastic when it is evaluated on states beyond the dataset. In this work, we use Euclidean distance to evaluate this notion of similarity. To build such a diffusion term, we propose a novel training objective that encourages the entries in the diffusion matrix to have local minima at each of the training data points, and to be strongly convex in a small surrounding neighborhood, while simultaneously ensuring that the neural SDE's predictions fit the training data.

By leveraging the physics-based knowledge encoded in the drift term's structure, the proposed models are capable of making accurate predictions of the dynamics, even when they are trained on limited amounts of data and evaluated on inputs well beyond the regime observed during data collection. Furthermore, the proposed diffusion term not only allows the models to represent stochastic dynamics but also acts as a conservative estimate of the epistemic uncertainty in the model's predictions. Finally, the proposed neural SDEs can be evaluated quickly enough for use in online model-predictive control algorithms, or they can be used as offline simulators for model-based reinforcement learning.

We demonstrate these capabilities through experiments on simulated robotic systems, as well as by using them to model and control a hexacopter's flight dynamics (illustrated in Figure 1). The prediction accuracy of the proposed neural SDEs is an order of magnitude higher than that of a baseline model, and the neural SDEs also more reliably estimate the uncertainty in their own predictions. When applied to an offline model-based RL task, the control policies that result from our neural SDEs achieve similar performance to model-free RL policies, while requiring $30\times$ less data.

**Related work.** Neural ordinary differential equations (ODEs) [6], which parametrize the right-hand side of a differential equation using a neural network, are a class of models that provide a natural mechanism for incorporating existing physics and engineering knowledge into neural networks [1, 7]. For example, many works use the Lagrangian, Hamiltonian, or Port-Hamiltonian formulation of dynamics to inform the structure of a neural ODE [8, 9, 10, 11, 12, 13, 14, 15, 16, 17, 18, 19, 20]. Other works use similar physics-inspired network architectures to learn models that are useful for control [21, 22, 23, 24, 25, 26, 27, 28, 29]. However, these models are typically deterministic and cannot estimate their own uncertainty. By contrast, designing uncertainty-aware models is a primary focus of this work. On the other hand, neural SDEs have also been recently studied in the context of generative modeling, learning stochastic dynamics, and estimating the uncertainty in neural network parameters [30, 31, 32, 33, 34, 35, 36]. However, to the best of our knowledge, we are the first to propose the use of neural SDEs to capture model uncertainty while also leveraging a priori physics knowledge, and to apply them for model-based RL and control of dynamical systems.

Estimates of model uncertainty are crucial to preventing *model exploitation* in model-based reinforcement learning and control, particularly in the offline setting [2, 3, 4, 5, 37]. Gaussian process regression has been successfully used to learn probabilistic models of system dynamics or of additive residual terms [2, 38, 39, 40]. However, it can be challenging to scale such methods to large datasets and high-dimensional systems, or to account for general non-additive interactions between the known and unknown portions of the dynamics. Meanwhile, many model-based RL algorithms use neural networks to parametrize the system dynamics. Such algorithms typically also rely on measures of model uncertainty, or alternatively on some measure of the *distance* to the previously observed training data, during policy synthesis [41, 42, 3, 43, 4, 44, 45, 46, 47, 48]. One popular method for uncertainty estimation is *probabilistic ensembles* [3], which uses ensembles of Gaussian distributions (whose mean and variance are parametrized by neural networks) to represent aleatoric and epistemic uncertainty. We similarly propose a method to learn uncertainty-aware models of dynamical systems. However, in contrast with existing ensemble-based methods, the proposed neural SDEs are capable of leveraging a priori physics knowledge, they only require the training and memory costs associated with a single model, and they provide uncertainty estimates that are explicitly related to the distance of the evaluation point to the training dataset.

## 2 Our Approach: Physics-Constrained, Uncertainty-Aware Neural SDEs

Throughout the paper, we assume that a dataset $\mathcal{D} = \{\tau_1, \ldots, \tau_{|\mathcal{D}|}\}$ of system trajectories is available in lieu of the system's model, where $\tau = \{(x_0, u_0), \ldots, (x_{|\tau|}, u_{|\tau|})\}$, $x_i \in \mathcal{X}$ is the state of the system at a time $t_i(\tau)$, $u_i \in \mathcal{U}$ is the state-dependent control input applied at time $t_i(\tau)$, and $\mathcal{X}, \mathcal{U}$ are the state and input spaces, respectively. Given any current state value $x_k$ at time $t_k$, and any control signal $u$ or sequence of control inputs, we seek to learn a model to predict trajectories $x_{k+1}, \ldots, x_{k+n_r}$ of length $n_r$. Specifically, our objective is to train neural network models of dynamical systems that account for stochastic dynamics, that explicitly model prediction uncertainties, and that allow for the incorporation of a priori physics-based knowledge.

### 2.1 Using Neural SDEs to Model Systems with Unknown Dynamics

We propose to use neural SDEs to learn models of unknown dynamics, as opposed to learning ensembles of probabilistic models that directly map states and control inputs to distributions over future states [49, 3]. A neural SDE is an SDE whose drift and diffusion terms are parametrized by neural networks, and its state is a stochastic process that evolves according to

$$\mathrm{d}x = f_\theta(x, u)\,\mathrm{d}t + \Sigma_\phi(x, u) \star \mathrm{d}W, \tag{1}$$

where $f_\theta : \mathcal{X} \times \mathcal{U} \to \mathbb{R}^{n_x}$ and $\Sigma_\phi : \mathcal{X} \times \mathcal{U} \to \mathbb{R}^{n_x \times n_w}$ are the drift and diffusion terms, respectively, and $W$ is the $n_w$-dimensional Wiener process. The notation $\star$ indicates that the SDE is either in the Itô [50] or Stratonovich [51] form. The reader unfamiliar with these forms should feel free to ignore the distinction [52, 53, 30], which becomes an arbitrary modeling choice when $f_\theta$ and $\Sigma_\phi$ are learned. Under mild Lipschitz assumptions [54, 55] on $f_\theta, \Sigma_\phi$, we can efficiently sample a distribution of predicted trajectories via numerical integration of the neural SDE [56, 57]. Furthermore, we can incorporate physics knowledge into such models, discussed further in Section 2.3.

**Efficient SDE Training.** Given $\mathcal{D}$, we train the unknown functions $f_\theta$ and $\Sigma_\phi$ in an end-to-end fashion. For fixed $\theta$ and $\phi$, we integrate the neural SDE (1) to obtain a set of system trajectories over

the time horizon $[t_i(\tau), t_{i+n_r}(\tau)]$. We then define the training objective

$$\mathcal{J}_{\text{data}} = \sum_{(x_i, u_i, \ldots, x_{i+n_r}) \in \mathcal{D}} \frac{1}{n_p} \sum_{j=i+1}^{n_r} \sum_{p=1}^{n_p} (x_j^p - x_j)^{\text{T}} S^{-1} (x_j^p - x_j), \qquad (2)$$

$$\{x_{i+1}^p, \ldots, x_{i+n_r}^p\}_{p=1}^{n_p} = \text{SDESolve}(x_i, u_i, \ldots, u_{i+n_r-1}; f_\theta, \Sigma_\phi),$$

where $(x_i, u_i, \ldots, x_{i+n_r}) \in \mathcal{D}$ denotes any sequence of transitions of length $n_r$ from the dataset, the predicted trajectory $x_{i+1}^p, \ldots, x_{i+n_r}^p$ is the $p$-th path sampled by any *differentiable* numerical integrator SDESolve, and $n_p$ is the total number of sampled paths. We optimize the parameters $\theta$ and $\phi$ by minimizing $\mathcal{J}_{\text{data}}$ using either automatic differentiation through SDESolve, or via the adjoint sensitivity method for SDEs [33].

For a given time index $j$, the summand in (2) is proportional to the negative log-likelihood of the observed state $x_j$ with respect to a fixed-variance Gaussian distribution with covariance matrix $S$ centered around the predicted state $x_j^p$. Minimizing $\mathcal{J}_{\text{data}}$ is thus equivalent to minimizing the likelihood of the observed trajectories under the neural SDE, given a Gaussian state observation model. This formulation has been a common choice in the literature [33, 30], and is suitable for many robotics and engineering settings. However, we expect that more nuanced forms of aleatoric uncertainty can be captured by adapting $\mathcal{J}_{\text{data}}$, e.g., using kernel density estimation to approximate the SDE-generated distribution over future states when defining the data likelihood.

## 2.2 Using Neural SDEs to Represent Distance-Aware Estimates of Uncertainty

We emphasize that direct optimization of the model as presented in Section 2.1 would result in the diffusion approaching zero, if the drift term alone could be used to fit the training data well [31]. Instead, we propose to design, constrain, and train the diffusion term to represent modeling uncertainties. In particular, we propose an approach to learn a *distance-aware* diffusion term $\Sigma_\phi^{\text{dad}}$ that may capture the true stochasticity of the dynamics when it is evaluated near points in the training dataset (in terms of Euclidean distance), while constrained to output highly stochastic predictions for points that are far from the dataset. A feature of our approach is that $\Sigma_\phi^{\text{dad}}$ is a neural network trained by locally sampling points near the training dataset, and yet its predictions respect useful global properties. Specifically, we translate this notion of distance-awareness into several mathematical properties that $\Sigma_\phi^{\text{dad}}$ must satisfy and propose a suitable loss function for its training. For notational simplicity, we use $[\text{xu}]_i$ to denote the vector that results from concatenating $x_i$ and $u_i$.

**Bounded diffusion far from training dataset.** On evaluation points sufficiently far from the training dataset, the entries in the diffusion term should saturate at maximum values. We accordingly propose to write $\Sigma_\phi^{\text{dad}}([\text{xu}]) := \Sigma^{\max}([\text{xu}]) \odot h_\phi(\eta_\psi([\text{xu}]))$, where $\eta_\psi : \mathcal{X} \times \mathcal{U} \to [0, 1]$ is a parametrized scalar-valued function that encodes the notion of distance awareness, $h_\phi : [0, 1] \to [0, 1]^{n_x \times n_w}$ is a parameterized element-wise monotonic function, $\Sigma^{\max} : \mathcal{X} \times \mathcal{U} \to \mathbb{R}^{n_x \times n_w}$ is a matrix specifying element-wise maximum values, and $\odot$ denotes the element-wise product.

**Small diffusion values near the training dataset.** On points included in the training dataset, $\Sigma_\phi^{\text{dad}}$ should not introduce additional stochasticity related to epistemic uncertainty. We ensure that the model's outputs are less stochastic when evaluated on points near the training dataset than on those far from it, by requiring that each entry of the diffusion term has a local minima at each pair $(x_i, u_i)$ in $\mathcal{D}$. We impose this property on $\Sigma_\phi^{\text{dad}}$ by training $\eta_\psi$ to have zero-gradient values at every point in the training dataset. This yields the loss function $\mathcal{J}_{\text{grad}} = \sum_{[\text{xu}]_i \in \mathcal{D}} \left\| \nabla_{[\text{xu}]} \eta_\psi([\text{xu}]_i) \right\|$.

**Increasing diffusion values along paths that move away from the training dataset.** As the query point moves away from the points in the training dataset, the magnitude of the diffusion term should monotonically increase. Let $\Gamma$ be any path along which the distance from the current point to the nearest training datapoint always increases. Then, along $\Gamma$, the entries of $\Sigma_\phi^{\text{dad}}$ should monotonically increase. We enforce this property via local strong convexity constraints near the training dataset. Specifically, for every datapoint $(x_i, u_i) \in \mathcal{D}$ and a fixed radius $r > 0$, we design $\eta_\psi$ to be strongly convex within a ball $\mathcal{B}_r(x_i, u_i) := \{(x, u) \mid \|[\text{xu}] - [\text{xu}]_i\| \leq r\}$ with a strong convexity constant $\mu_i > 0$. That is, for any $(x, u), (x', u') \in \mathcal{B}_r(x_i, u_i)$, we desire that the constraint $\text{SC}_\psi(x, u, x', u') \geq 0$ holds, where $\text{SC}_\psi$ is defined as:

$$\text{SC}_\psi(x, u, x', u') := \eta_\psi([\text{xu}]') - \eta_\psi([\text{xu}]) - \nabla_{[\text{xu}]} \eta_\psi([\text{xu}])^\top ([\text{xu}]' - [\text{xu}]) - \mu_i \|[\text{xu}]' - [\text{xu}]\|^2.$$

Now, to eliminate the strong convexity constants $\mu_i$ as tuneable hyperparameters, we parametrize a function $\mu_\zeta : \mathcal{X} \times \mathcal{U} \to \mathbb{R}_+$ using a neural network. Note that while $\mu_\zeta$ is a continuous function over $\mathcal{X} \times \mathcal{U}$, we only ever evaluate it at points in the training dataset. This is effectively equivalent to learning separate values of $\mu_i$ for every $(x_i, u_i) \in \mathcal{D}$. We then define the loss functions

$$\mathcal{J}_{\text{sc}} = \sum_{(x_i, u_i) \in \mathcal{D}} \sum_{\substack{(x, u), (x', u') \\ \sim \mathcal{N}((x_i, u_i), r)}} \begin{cases} 0, & \text{if } \text{SC}_\psi(x, u, x', u') \geq 0 \\ (\text{SC}_\psi(x, u, x', u'))^2, & \text{otherwise} \end{cases} \quad \text{and } \mathcal{J}_\mu = \sum_{(x_i, u_i) \in \mathcal{D}} \frac{1}{\mu_\zeta(x_i, u_i)},$$

where $\mathcal{N}((x_i, u_i), r)$ is a Gaussian distribution with mean $(x_i, u_i)$ and standard deviation $r$. Here, $\mathcal{J}_\mu$ is a regularization loss term that encourages high values of $\mu_i$. This ensures that $\eta_\psi$ reaches its maximum value of 1 close to the boundaries of the training dataset.

**Modeling details.** In practice, $\eta_\psi$ has a sigmoid at its output, $\eta_\psi(x, u) := \text{sigmoid}(\text{NN}_\psi(x, u))$. Furthermore, we define $h_\phi(z) := \text{sigmoid}(W \text{sigmoid}^{-1}(z) + b)$, where $W, b \in \mathbb{R}^{n_x, n_w}$, and we restrict each entry of $W$ to be greater than 1. Thus, $h_\phi(\eta_\psi(x, u)) = \text{sigmoid}(\text{NN}_\psi(x, u)W + b)$, which allows it to learn heterogeneous distributions of noise from the output of $\eta_\psi$.

**Empirical illustration of $\eta_\psi$.** To demonstrate the effectiveness of the designed constraints and loss functions, we train $\eta_\psi$ to optimize $\lambda_{\text{grad}} \mathcal{J}_{\text{grad}} + \lambda_{\text{sc}} \mathcal{J}_{\text{sc}} + \lambda_\mu \mathcal{J}_\mu$ on two illustrative datasets of 2D trajectories. We parametrize $\eta_\psi$ and $\mu_\zeta$ as neural networks with swish activation functions, 2 hidden layers of size 32 each for $\eta_\psi$ and size 8 each for $\mu_\zeta$. We pick the values $\lambda_{\text{grad}} = \lambda_{\text{sc}} = \lambda_\mu = 1$, and the ball radius $r = 0.05$. Figure 3 illustrates the learned $\eta_\psi$

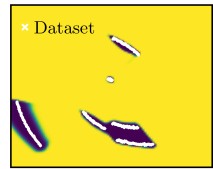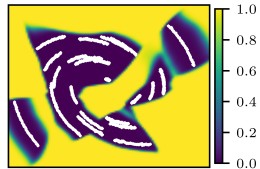

Figure 3: Output values of $\eta_\psi$ after training on illustrative sample datasets in a 2D domain.

evaluated on the grid $[-0.2, 0.2] \times [-0.2, 0.2]$. We observe that $\eta_\psi$ outputs low values near the training dataset, that these values increase as we move away from the dataset, and that the values are close to 1 when evaluated far from the dataset.

## 2.3 Leveraging A Priori Physics Knowledge in Neural SDEs

We represent the drift term $f_\theta$ in (1) as the composition of a known function – derived from a priori knowledge – and a collection of unknown functions that must be learned from data. That is, we write $f_\theta(x, u) := F(x, u, g_{\theta_1}(\cdot), \ldots, g_{\theta_d}(\cdot))$, where $F$ is a known differentiable function and $g_{\theta_1}(\cdot), \ldots, g_{\theta_d}(\cdot)$ are unknown terms within the underlying model. The inputs to these functions could themselves be arbitrary functions of the states and control inputs, or the outputs of the other unknown terms. Furthermore, by designing the element-wise maximum values $\Sigma^{\text{max}}(x, u)$ for the diffusion term, we can encode prior notions of uncertainty into the neural SDE by specifying the maximum amount of stochasticity that the model will predict when evaluated on any given region of the state space. We provide several examples of such physics-based knowledge in Section 3. The complete neural SDE learning problem can now be formulated as

$$\underset{\theta_1, \ldots, \theta_d, \psi, \phi, \zeta}{\text{minimize}} \quad \lambda_{\text{data}} \mathcal{J}_{\text{data}} + \lambda_{\text{grad}} \mathcal{J}_{\text{grad}} + \lambda_{\text{sc}} \mathcal{J}_{\text{sc}} + \lambda_\mu \mathcal{J}_\mu. \tag{3}$$

# 3 Experimental Results

We evaluate our approach by comparing its performance for modeling and model-based control tasks against state-of-the-art techniques such as probabilistic ensembles [49, 3, 58], system identification-based algorithms [59, 60, 61], and neural ODEs [6, 1]. Experimental details and additional results are available in the appendix. Code is available at https://github.com/wuwushrek/sde4mbrl.

## 3.1 Spring-Mass-Damper: Generalization and Uncertainty Estimation with Limited Data

We start with the problem of modeling the dynamics of an uncontrolled spring-mass-damper system given 5 *noisy trajectories* of 5 seconds each. The equations of motion are given by the state $x = [q, \dot{q}]$ where $q$ is the position of the mass and $\dot{q}$ is its velocity.

**SDE model.** We train the physics-informed model $\mathrm{d}x = [\dot{q}, g_{\theta_1}(x)]\mathrm{d}t + \sigma^{\max} \odot h_\phi(\eta_\psi(x)) \star \mathrm{d}W$, where the vector $\sigma^{\max}$ specifies the maximum level of stochasticity outside of the training dataset. We note that while this SDE model assumes $\ddot{q}$ is unknown, it knows that the rate of change of $q$ is $\dot{q}$.

**Neural SDE improves uncertainty estimates and accuracy over probabilistic ensembles.** As baselines for comparison, we train a neural ODE (which has the same architecture as the SDE model but excludes the diffusion term), and an ensemble of 5 probabilistic (Gaussian) models [58].

Figure 4 shows the prediction accuracy and uncertainty estimates of the learned models on a grid discretization of the state space. We use each point of the grid as an initial state for the models to predict 100 trajectories over a time horizon of 0.2 seconds. The first row shows that neural SDE and ODE models have high prediction accuracy. The neural SDE's accuracy decreases only slightly outside of the training dataset. On the other hand, for this low data regime, the probabilistic ensemble model is at

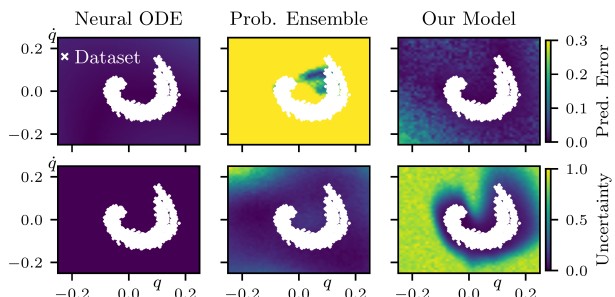

Figure 4: Model prediction errors and uncertainty estimates, when trained on the illustrated dataset.

least one order of magnitude less accurate than the neural SDE at points near the training dataset. The second row illustrates the variance of the model predictions, which we interpret as a measure of uncertainty. While the neural ODE has no notion of prediction uncertainty, the proposed neural SDE yields uncertainty estimates that agree with the availability of the training data. There is no clear relationship between the training dataset and the probabilistic ensemble's uncertainty predictions.

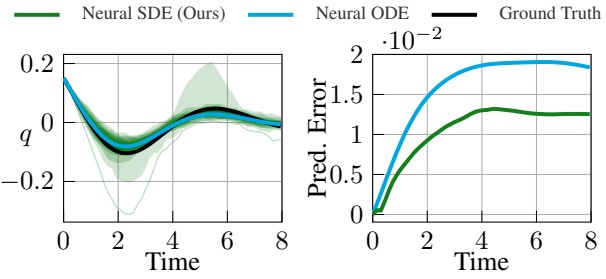

Figure 5: Neural SDE improves accuracy over neural ODE.

**Neural SDE generalizes beyond the training dataset.** Figure 5 shows the state trajectory predicted by the neural SDE and the neural ODE, from a given initial state. We observe that both models generalize well beyond the training dataset and are suitable for long-term prediction. However, the neural SDE's predictions are slightly more accurate than those of the neural ODE.

### 3.2 Cartpole Swingup: Offline Model-Based Reinforcement Learning with Neural SDEs

We now consider the cartpole swingup problem. The system state is defined by $x \coloneqq [p_x, \dot{p}_x, \theta, \dot{\theta}]$ and the scalar control input $u$. Here, $p_x$ is the position of the cart, $\theta$ is the angle of the pendulum, and $u$ is the force applied to the cart.

**Data collection.** In contrast to the Mass-Spring-Damper experiments, we now utilize a moderate amount of data to ensure that the probabilistic ensemble methods are able to learn reasonably accurate models of the dynamics. Specifically, we build two training datasets, each of which consists of 100 trajectories of 200 system interactions. The first dataset, referred to as *random dataset*, is collected by applying random control inputs. The second dataset, referred to as *on-policy dataset*, is collected by applying inputs generated from a policy trained through Proximal Policy Optimization (PPO) [62] and $600,000$ interactions with the ground truth dynamics model.

**Experimental setup.** We consider two physics-constrained neural SDE models with different prior knowledge. The first model, simply referred to as neural SDE, is given by $\mathrm{d}x = [\dot{p}_x, g_{\theta_1}(x, u), \dot{\theta}, g_{\theta_2}(x, u)]\mathrm{d}t + \sigma^{\max} \odot h_\phi(\eta_\psi(x)) \star \mathrm{d}W$ while the second model, referred to as neural SDE with side info, exploits the known control-affine structure of the dynamics and is given by: $\mathrm{d}x = [\dot{p}_x, g_{\theta_1}(x) + g_{\theta_2}(\theta, \dot{\theta})u, \dot{\theta}, g_{\theta_3}(x) + g_{\theta_4}(\theta, \dot{\theta})u]\mathrm{d}t + \sigma^{\max} \odot h_\phi(\eta_\psi(x)) \star \mathrm{d}W$.

We train both neural SDE models, neural ODE models with access to the same physics information, and an ensemble of probabilistic (Gaussian) models [58], using both the random and the on-policy

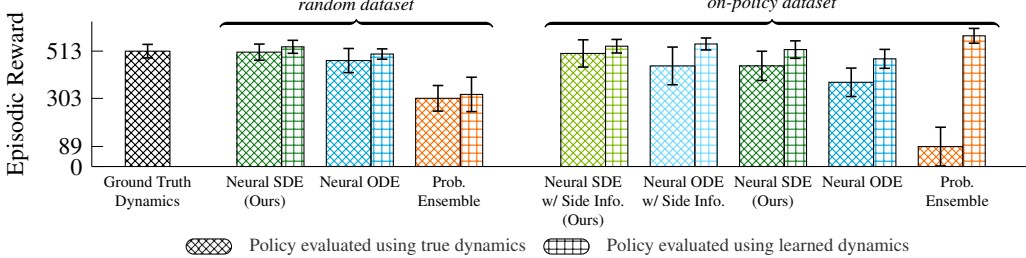

Figure 6: Reward of RL policies trained using learned dynamics.

datasets. For each learned model, we train a control policy via the PPO algorithm while using the model as an environment simulator. We compare the performance of the learned policies with the baseline policy used to generate the on-policy dataset. We note that the learned policies are trained on a dataset that is $30\times$ smaller than that required to train this baseline model-free RL policy. Figure 6 illustrates the mean episodic reward values achieved when evaluating the learned policies using the true dynamics (hatch pattern) vs using the learned dynamics models (grid pattern). The RL policies trained using the proposed neural SDE models consistently achieve higher rewards than policies trained using neural ODEs or probabilistic ensemble models of the dynamics.

**Offline model-based RL using neural SDEs is as performant as model-free RL, while requiring $30\times$ fewer interactions with the environment.** For the *random dataset*, the model-based policy trained using the neural SDE achieves comparable performance to the model-free baseline policy. For the *on-policy datset* (which is more limited in its coverage of the state-action space), the neural SDE results in a model-based policy that is less performant than the baseline policy. However, this gap is closed when considering the neural SDE that incorporates more physics knowledge.

**Neural SDEs are robust against model exploitation.** For the *on-policy dataset*, the probabilistic ensemble results in a model-based policy that performs much worse when operating in the true environment than it does when using the learned model as a simulator. However, the neural SDE appears to be much more robust to this symptom of model exploitation. We note that many model-based RL algorithms are designed explicitly to prevent the policy optimization procedure from exploiting model inaccuracies. We did not employ such algorithms in this experiment but instead used an implementation of the PPO algorithm with default parameters [63].

### 3.3 Hexacopter System: Real-Time, Learning-Based Control with 3 Minutes of Data

We now demonstrate that by using only basic knowledge of rigid-body dynamics and 3 minutes of flight data, our approach bridges the sim-to-real gap and enables real-time control of a hexacopter.

**Experimental setup.** Our hexacopter uses a *CubePilot Orange* flight controller running PX4 [64], and a *Beelink MINIS 12* as the main computational unit. The CubePilot fuses measurements from the onboard IMU and motion capture system to estimate the state $x := [p_W, v_W, q_{WB}, \omega_B]$ at a frequency of 100 Hz. Here $p_W$ and $v_W$ are the position and velocity in the world frame, $q_{WB}$ is the quaternion representing the body orientation, and $\omega_B$ is the angular rate in the body frame. The Beelink unit compiles our SDE models, solves a model predictive control (MPC) problem at each new state estimate, and sends back the resulting motor and desired angular rate commands.

**Data collection: 3 minutes worth of data.** We collect 3 system trajectories by manually flying the hexacopter via a radio-based remote controller. We store the estimated states $x$ and input motor commands $u \in [0, 1]^6$ at a frequency of 100 Hz. We obtain a total of 203 seconds worth of flight data. An analysis shows that 95% of the data corresponds to the hexacopter operating below the speed of 1.71m/s, absolute roll of 18°, and absolute pitch of 13° while the maximum absolute speed, roll, and pitch attained are respectively 2.7 m/s, 23°, and 23°.

**SDE model.** The physics-constrained model leverages the structure of 6-dof rigid body dynamics while parametrizing the aerodynamics forces and moments, the motor command to thrust function, and (geometric) parameters of the system such as the mass and the inertia matrix:

$$
\mathrm{d}\begin{bmatrix} v_W \\ \omega_B \end{bmatrix} = \begin{bmatrix} \frac{1}{m_\theta}\left(q_{WB}\left(T_\theta(u) + f_\theta^{\mathrm{res}}(x^{\mathrm{feat}})\right)\bar{q}_{WB}\right) + g_W \\ J_\theta^{-1}\left(M_\theta(u) + M_\theta^{\mathrm{res}}(x^{\mathrm{feat}})\right) - \omega_B \times J_\theta\omega_B \end{bmatrix}\mathrm{d}t + \sigma^{\max} \odot h_\phi(\eta_\psi(x^{\mathrm{feat}})) \star \mathrm{d}W,
$$

where $x^{\text{feat}} = [v_W, \omega_B]$, $\times$ denotes the cross product, $\bar{q}_{WB}$ is the conjugate of $q_{WB}$, the vector $\sigma^{\max}$ is the maximum diffusion calibrated by overestimating sensor noise, the variables $m_\theta$ and $J_\theta = \text{diag}(J_\theta^{\text{x}}, J_\theta^{\text{y}}, J_\theta^{\text{z}})$ represent the system mass and inertia matrix, the neural network functions $f_\theta^{\text{res}}$ and $M_\theta^{\text{res}}$ represent the residual forces and moments due to unmodelled and higher order aerodynamic effects, the parametrized functions $T_\theta$ and $M_\theta$ estimate the map between motor commands, thrusts, and moment values. We use $g_W$ to denote the gravity vector. Our model assumes the elementary knowledge that $\mathrm{d}p_W = v_W \mathrm{d}t$ and $\mathrm{d}q_{WB} = 0.5 q_{WB}[0, \omega_B]\mathrm{d}t$, which completes the full SDE model.

**Neural SDE improves prediction accuracy over system identification while also providing uncertainty estimates.** As a baseline model, we remove the diffusion term in our SDE model and use a nonlinear system identification (SysID) [61, 60] algorithm to learn the parametrized terms $m_\theta$, $J_\theta$, $f_\theta^{\text{res}}$, $M_\theta^{\text{res}}$, $T_\theta$ and $M_\theta$. The nonlinear unknown functions are parametrized using a polynomial basis. Figure 7 shows the model predictions along a trajectory that was not included in

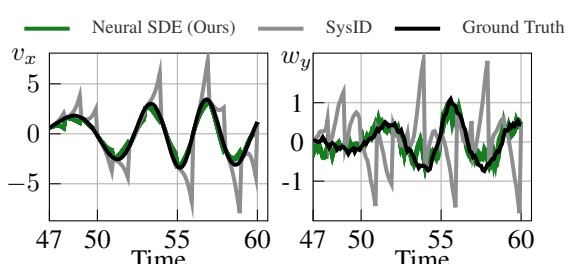

Figure 7: Our neural SDE achieves accurate and stable long-horizon predictions.

the training dataset. We split the trajectory into segments of 1 second and perform open-loop prediction of the state evolution given the segment's initial state and the motor commands. The proposed neural SDE demonstrates improved accuracy and stability in comparison with the SysID baseline. We include additional comparisons to neural ODE and SysID baselines for the prediction and control of a high-fidelity quadcopter simulation in the appendix.

**Neural SDE results in highly performant MPC even when operating beyond the training regime.** Figure 8 shows the result of using the learned neural SDE in an MPC algorithm to track a lemniscate trajectory. The trajectory is obtained through generic minimum snap trajectory generation [65] without any knowledge of the hexacopter dynamics. Figure 8 demonstrates the ability of the neural SDE to generalize far beyond the limited training dataset, and to achieve high performance in terms of tracking accuracy when used for MPC. For example, for the first 10 seconds, the hexacopter reaches speed up to 1.7 m/s, yet the tracking error remains only 6 cm. We provide additional results in the supplementary material for tracking an aggressive circle trajectory.

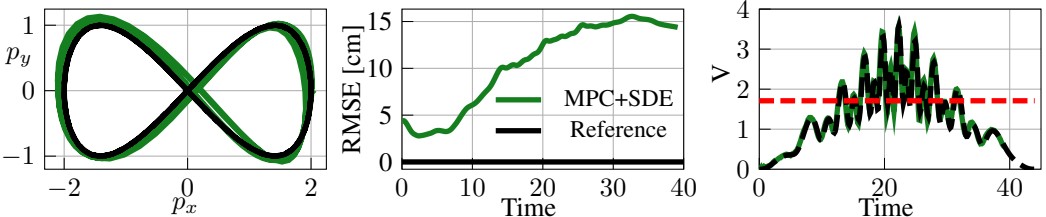

Figure 8: Tracking performance on the lemniscate trajectory. We obtain a root mean squared error (RMSE) of 15 cm despite operating at speeds up to 3.6 m/s and roll, pitch angles up to 32°, nearly twice as high as the maximum values from the training dataset (shown by the dashed red line).

## 4 Discussion and Limitations

We present a framework and algorithms for learning uncertainty-aware models of controlled dynamical systems, that leverage a priori physics knowledge. Through experiments on a hexacopter and on other simulated robotic systems, we demonstrate that the proposed approach yields data-efficient models that generalize beyond the training dataset, and that these learned models result in performant model-based reinforcement learning and model predictive control policies.

While our formulation is general enough to extend to partial state observations easily, our current experiments use noisy observations of the system state. Future work will extend the proposed approach to learn dynamics from more varied observations of the state, e.g., from image observations. Furthermore, future work will explore methods to improve the neural SDE's ability to capture nuanced forms of aleatoric uncertainty, e.g., using kernel density estimation to approximate the SDE's state distribution when defining the loss function.

**Acknowledgments**

This work was supported in part by AFOSR FA9550-19-1-0005 and NSF 2214939.

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

# How to Learn and Generalize From Three Minutes of Data: Physics-Constrained and Uncertainty-Aware Neural Stochastic Differential Equations

# Appendix

## A    Implementation and Modeling Details

We implement all the numerical experiments using the python library JAX [66], in order to take advantage of its automatic differentiation and just-in-time compilation features. We use Python 3.8.5 for the experiments and train all our models on a laptop computer with an Intel i9-9900 3.1 GHz CPU with 32 GB of RAM and a GeForce RTX 2060, TU106. We provide the code for all experiments at https://github.com/wuwushrek/sde4mbrl.

**Training optimizer hyperparameters.**    We use the *Adam* optimizer [67] for all optimization problems. This includes when training the neural ODE model, the probabilistic ensemble model, and the system identification-based model. We use the default hyperparameters for the optimizer, except for the learning rate, which we linearly decay from $0.01$ to $0.001$ over the first $10000$ gradient steps. We use early stopping criteria for all our experiments. We use a batch size of $512$ for the neural ODE, SDE models, and the system identification-based model. Instead, we use a batch size of $32$ for the ensembles of probabilistic models.

**Model design.**    We use the same batch size and learning rate scheduler for the optimizer, the same neural network architecture for $\eta_\psi, h_\phi$, and $\mu_\zeta$, and the same parameters $n_p, \lambda_{\text{data}}, \lambda_{\text{grad}}, \lambda_{\text{sc}}$. Only $r, \lambda_\mu$, and $\Sigma^{\max}$ vary across experiments. Specifically, we parametrize $\eta_\psi$ as a feedforward neural network with $\mathrm{swish}$ activation functions and 2 hidden layers of size $32$ each while the parameters $W, b$ of $h_\phi$ are matrices of size corresponding to the dimension of $\sigma^{\max}$. We parametrize $\mu_\zeta$ as a feedforward neural network with $\tanh$ activation functions and 2 hidden layers of size $8$ each. For the loss function penalty terms, we use $\lambda_{\text{data}} = 1.0$ for the data loss, $\lambda_{\text{grad}} = 0.01$ for the zero-gradient loss, and $\lambda_{\text{sc}} = 0.01$ for the strong convexity loss. We use $n_p = 1$ for training the neural SDE model. For the ensembles of probabilistic models, we always use 5 models in the ensemble, where each model is a Gaussian with mean and variance parametrized by feedforward neural networks with 2 hidden layers of size $64$ each and $\mathrm{silu}$ activation functions.

**Training times of neural ODEs and SDEs.**    For the Cartpole swing-up example, training a neural ODE model took 2 minutes and 4 seconds on average, with a standard deviation of 23 seconds over 5 training runs. In contrast, training our neural SDE model took 4 min and 30 seconds on average, with a standard deviation of 42 seconds over 5 training runs. We used the Euler integration scheme for the ODE and the Euler-Maruyama integration scheme for the SDE. While training neural SDEs, we use a single particle for all the examples in the paper as it reduces training times without affecting model performance.

**Inference times of neural ODEs and SDEs.**    For the Cartpole swing-up example, we compare inference times of neural ODE and neural SDE for a prediction horizon $n_r = 100$ and by repeating the simulation over 10 different initialization. Inference times for neural ODE was $0.16 \pm 0.05 ms$. Using 50 particles, the inference time of neural SDE is $1.11 \pm 0.1 ms$. For the hexacopter system, we also compare inference times of neural ODE and neural SDE for a prediction horizon $n_r = 30$ and by repeating the simulation over 10 different initialization. Inference times for neural ODE was $0.07 \pm 0.05 ms$. In contrast, using a single particle ($n_p = 1$), the inference time for neural SDE is $0.13 \pm 0.06 ms$. Using 50 particles, the inference time of neural SDE is $0.7 \pm 0.15 ms$.

# B  Supplementary Mass-Spring-Damper Details and Results

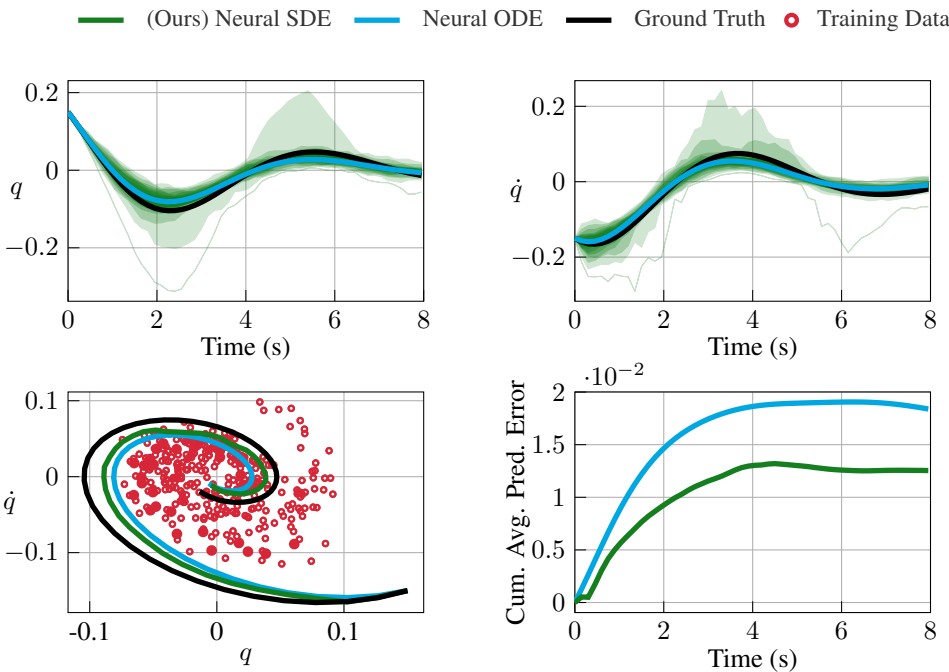

Figure 9: Prediction of the neural SDE and neural ODE models over a time horizon of 8 seconds for an initial condition $x_{\text{init}} = [0.15, -0.15]$ outside of the training dataset. The neural SDE generalizes well beyond the training dataset while providing accurate coverage of the groundtruth trajectory and slightly improving accuracy over the neural ODE model.

The equations of motion are given by the state $x := [x_1, x_2] = [q, \dot{q}]$ with $\dot{x}_1 = x_2$ and $m\dot{x}_2 = -bx_2 - kx_1$, where $q$ is the position of the mass, $m$ is the mass, $b$ is the damping coefficient, and $k$ is the spring constant. We consider the case where $m = 1$, $b = 0.5$, and $k = 1$; all the quantities being in the international system of units.

**Data collection: Noisy and extremely scarce amount of data.**  We collect two dataset $\mathcal{D}_1$ and $\mathcal{D}_2$ of 5 *trajectories* each. The trajectories are obtained from the known dynamics with the initial positions randomly sampled in the top right quadrant $[0.1, 0.1] \times [0.05, 0.15]$ for $\mathcal{D}_1$ and in the more broad region $[-0.1, 0.1] \times [-0.1, 0.1]$ for $\mathcal{D}_2$. Each trajectory has a length of 5 seconds and is integrated through the Euler method with a discrete step size of 0.01 second. Besides, we add a zero-mean Gaussian noise with a standard deviation of $[0.005, 0.01]$ to each state measurement.

Specifically, we use the first dataset to show that our neural SDE framework provides interpretable uncertainty estimates and better prediction accuracy than Gaussian ensemble models in the extremely low and non-diverse data regime. The second dataset will be used to show that our neural SDE model improves prediction accuracy over neural ODE when the dataset is sufficiently diverse, even in the low data regime.

**Benchmark models.**  We assume that the dynamics of the mass-spring-system are unknown, and we use the proposed neural SDE framework to train predictive models from the training dataset $\mathcal{D}_1$ and $\mathcal{D}_2$. Our neural SDE model has the following structure:

$$\mathrm{d}x = [x_2, g_\theta(x)]\mathrm{d}t + \sigma^{\text{max}} \odot h_\phi(\eta_\psi(x)) \star \mathrm{d}W,$$

where $g_\theta$ is a feedforward neural network with $\tanh$ activation functions and 2 hidden layers of size 4 and 16, respectively. The vector $\sigma^{\text{max}} := [\sigma_1^{\text{max}}, \sigma_2^{\text{max}}] = [0.001, 0.02]$ provides the desired diffusion values outside of the training dataset as formally defined in Section 2.2.

We compare the neural SDE models with neural ODE and ensemble of probabilistic (Gaussian) models. The neural ODE models are trained with the same architecture as the neural SDE models

without the diffusion term. We use Euler-Maruyama and Euler methods as the SDESolve algorithms, respectively, with a step size of $0.01$ and a horizon $n_{\mathrm{r}} = 50$ during training. Further, we use $\lambda_\mu = 1$ for encouraging large strong convexity coefficients and a ball radius $r = 0.05$ to enforce the strong convexity property locally via sampling in the neighborhood of the dataset.

**Neural SDE generalizes beyond the training dataset.** Figure 5 shows the state evolution of the neural SDE and neural ODE models trained on the more diverse dataset $\mathcal{D}_2$ and evaluated for an initial state outside the training dataset. We can observe that the SDE generalizes well beyond the training dataset and is suitable for long-term prediction. On the plot representing the evolution of $\dot{q}$ as a function of time, we observe that the noise at the beginning is high as we start from a point outside of the training dataset. We emphasize that we do not show the trained Gaussian ensemble model on this plot as it diverges over $0.3$ seconds of integration time.

## C Supplementary Experimental Results for the Cartpole Swingup Task

**Benchmark models.** We use feedforward neural networks with $\texttt{tanh}$ activation functions and 2 hidden layers of size 8 and 24 respectively for $g_{\theta_1}$ and $g_{\theta_2}$ in the non-control-affine case. In the control-affine setting, we use feedforward neural networks with $\texttt{tanh}$ activation functions and 2 hidden layers of size 8 and 24 respectively for $g_{\theta_1}$ and $g_{\theta_3}$, while $g_{\phi_2}$ and $g_{\phi_4}$ are smaller networks with $\texttt{tanh}$ activation functions and 2 hidden layers of size 6 and 8, respectively. The vector $\sigma^{\max} = [0.005, 0.05, 0.004, 0.01]$ provides the desired diffusion values beyond the dataset coverage.

We compare the neural SDE models with neural ODE and ensemble of probabilistic (Gaussian) models. The neural ODE models are trained with the same architecture as the neural SDE models without the diffusion term. We use Euler-Maruyama and Euler methods as the SDESolve algorithms, respectively, with a step size of $0.02$ and a horizon $n_{\mathrm{r}} = 30$ during training. Further, we use $\lambda_\mu = 10$ for encouraging large strong convexity coefficients and a ball radius $r = 0.2$ to enforce the strong convexity property locally.

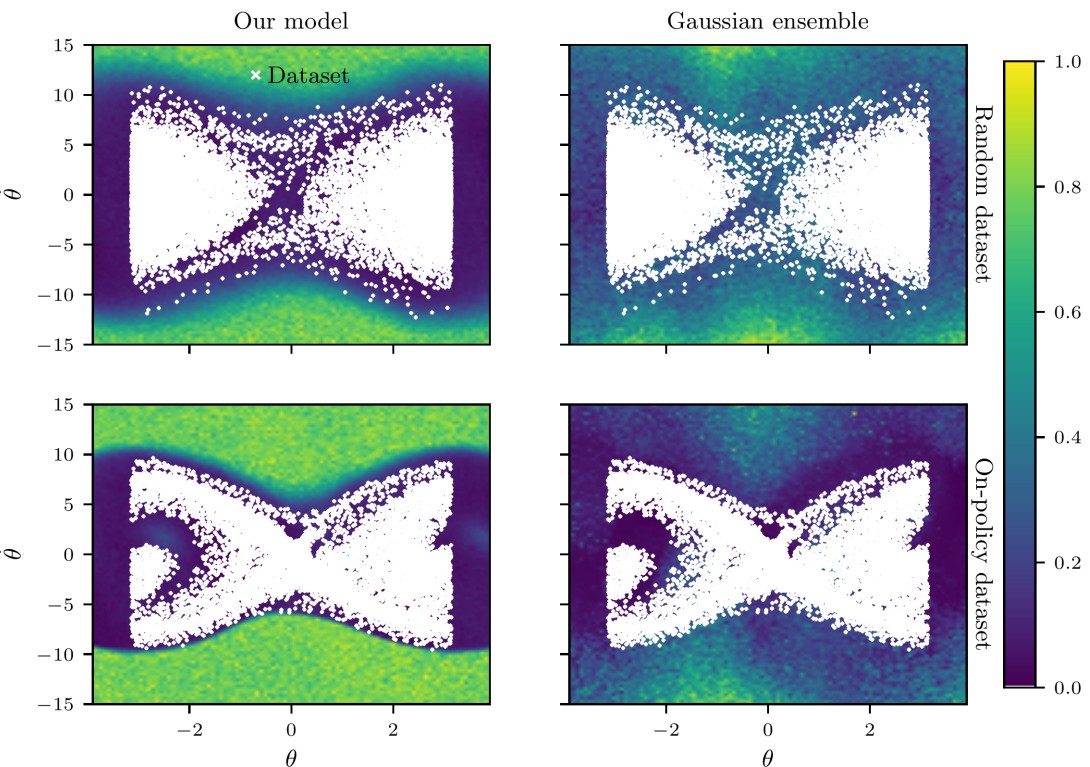

Figure 10: Uncertainty estimates of the neural SDE and Gaussian ensemble models on the on-policy and random datasets.

**Neural SDE provides distance-aware uncertainty estimates.** Figure 10 illustrates the uncertainty estimates of the neural SDE and probabilistic ensemble on both the on-policy and random datasets. The neural SDE model provides low uncertainty estimates around the region of the state space covered by the training dataset and high uncertainty estimates outside of the training dataset. Instead, although the probabilistic ensemble seems to provide high uncertainty estimates far away from the dataset, it often provides high uncertainty estimates around the dataset.

**Neural SDE improves prediction accuracy over the probabilistic ensemble.** Figure 11 illustrates the prediction accuracy of the neural SDE and probabilistic ensemble on trajectories of 1 second. We observe that, even with a moderate amount of data, the neural SDE provides more accurate trajectory predictions than probabilistic ensembles. This observation matches our results in the Mass-Spring-Damper experiments, where the neural SDE also provides more accurate pre-

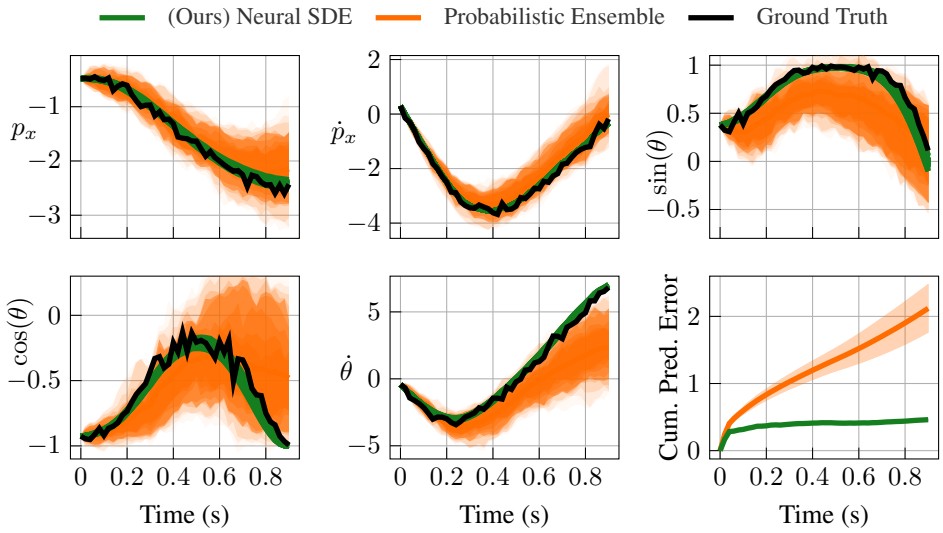

Figure 11: Comparison of the prediction accuracy of the neural SDE and probabilistic ensemble on trajectories of 1 second.

dictions than the probabilistic ensemble in the low data regime, while the probabilistic ensemble's predictions explode even for short horizon trajectories.

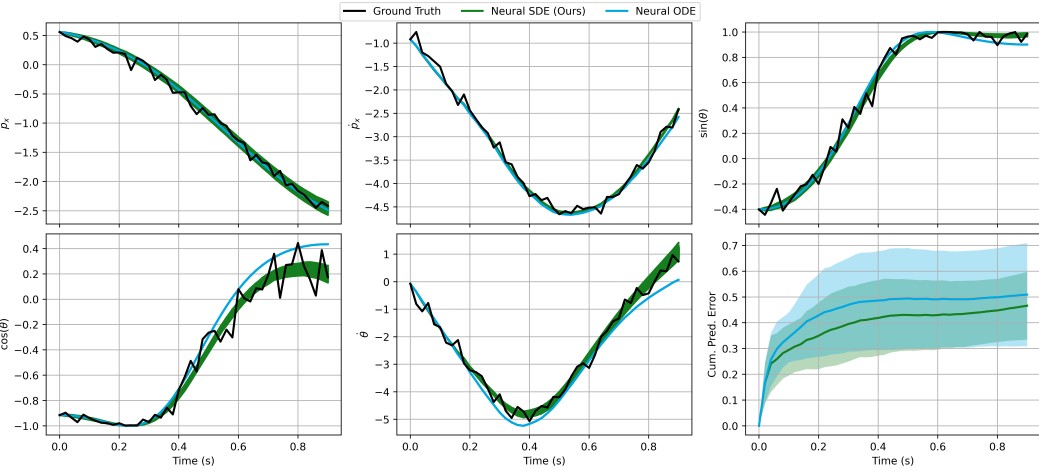

Figure 12: Model accuracy of black-box neural SDEs and neural ODEs on the Cartpole system when predicting trajectories *in* the training dataset.

**Neural SDEs yield models with improved robustness to data noise and generalization beyond the training dataset compared to neural ODEs.** Figure 12 and Figure 13 illustrate the prediction accuracy of the neural SDE and ODE models learned on the on-policy dataset when evaluated on randomly chosen initial states and control inputs inside and outside of the dataset, respectively. By inside of the training dataset, we mean that the initial state and the control actions are picked from the dataset. Instead, by outside of the training dataset, we mean trajectories generated by picking random actions in the environment as opposed to the suboptimal policy employed to generate the training dataset. The learned models are evaluated at initial states randomly picked either inside or outside the training dataset, and by applying the corresponding control inputs in an open loop manner. The accuracy plot (bottom and far-right figure) shows the prediction error of the learned models when evaluated on 25 initial state predictions. When evaluated near the training dataset, the

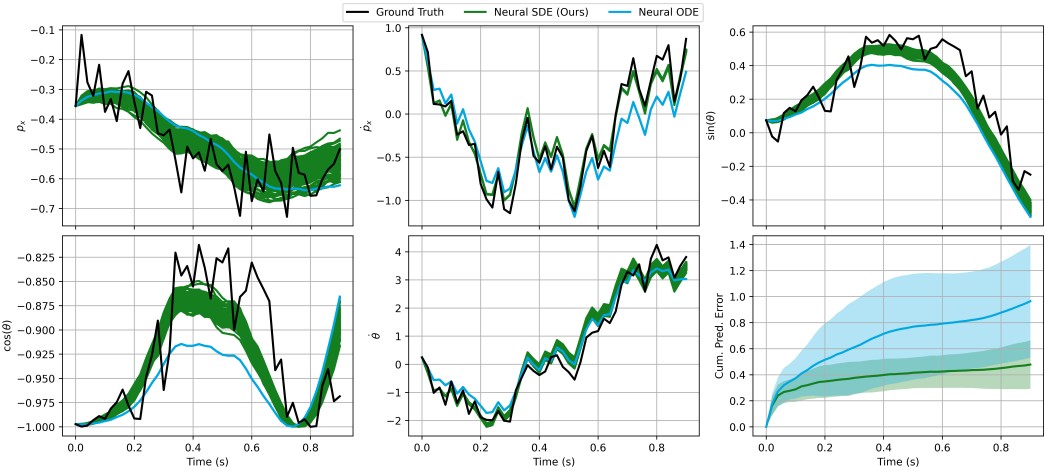

Figure 13: Model accuracy of black-box neural SDEs and neural ODEs on the Cartpole system when predicting trajectories *outside* of the training dataset.

neural SDE-based model slightly improves prediction accuracy over neural ODEs, while providing uncertainty estimates. The performance gap becomes larger when evaluating outside of the training dataset. Figure 13 shows that the neural SDE model improves accuracy over the neural ODE model when evaluated on unseen data. This figure demonstrates the generalization and robustness to noise of neural SDEs compared to neural ODEs. *This gap in the prediction accuracy could explain why the reinforcement learning policy trained using our neural SDE outperforms the neural ODE-based policy in Figure 6.*

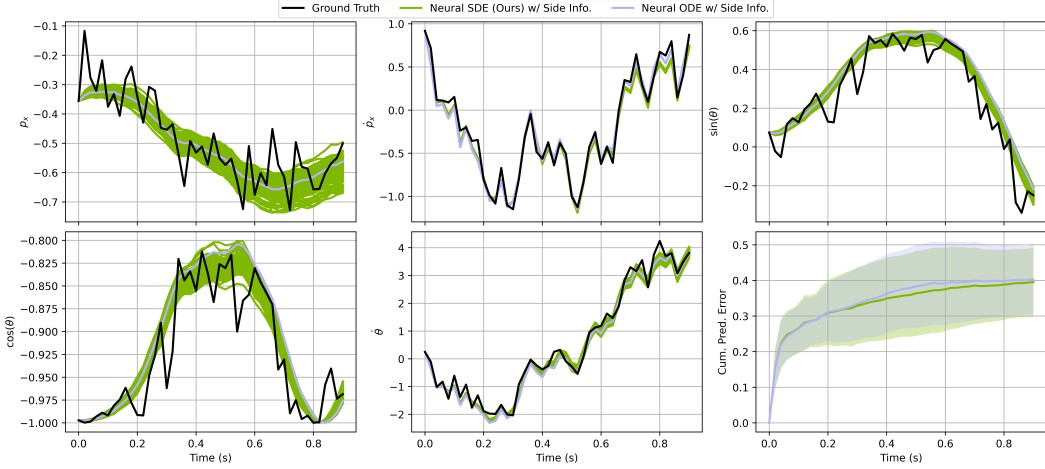

Figure 14: Model accuracy of neural SDEs and neural ODEs models with the control-affine side information when predicting trajectories *outside* of the training dataset.

**Uncertainty estimates matter for model-based control: With side information, our neural SDE-based policies outperform neural ODE-based policies despite the two models showing similar prediction accuracy.** With the control-affine dynamics side information, there is no gap in prediction accuracy between the neural SDE and ODE model when evaluating on the training dataset and outside of the training dataset. This contrasts the black-box neural SDE and ODE models in Figure 12 and 13, where the neural ODE model has a lower prediction accuracy. However, the policy trained using our neural SDE outperforms the neural ODE-based policy as illustrated in

Figure 6. This result emphasizes the need for uncertainty estimates in offline model-based reinforcement learning, as the neural SDE and ODE models used for training exhibit comparable predictive capabilities yet perform differently when doing control.

# D Supplementary Hexacopter Details and Results

This numerical experiment shows that with basic knowledge of rigid-body dynamics as prior physics knowledge for our neural SDE, we can learn accurate and uncertainty-aware predictive models for multirotor systems from only 3 minutes of manual flight. Then, using our learned model in a nonlinear model predictive control (NMPC) framework, we show incredible tracking performance on aggressive trajectories, despite how the reference trajectories push the system to operate far beyond what was seen during training. Besides, we provide additional results in Gazebo [68], a high-fidelity platform for software-in-the-loop simulation of multirotors, as a safe approach to compare the controllers based on the learned neural SDE, ODE, and system identification-based models.

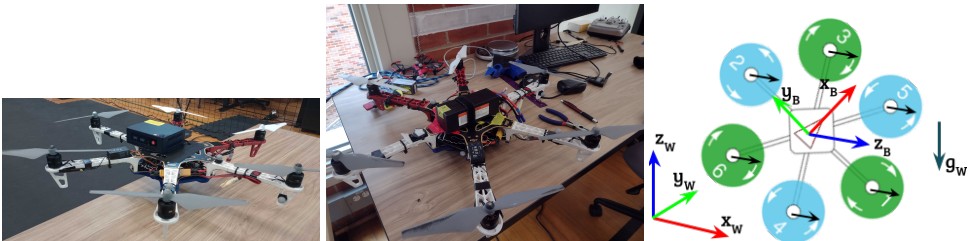

Figure 15: Our experimental Hexacopter and the frame used for modeling its dynamics.

**Experimental Setup.** In our hardware experiments, we use a custom-built hexacopter with *CubePilot Orange* as the flight controller running PX4 [64]. The hexacopter is equipped with 920KV brushless motors, 10 inch and two-bladed propellers, and it features the *DJI F550* frame, which has a 550mm diagonal motor to motor distance. In contrast, in our software-in-the-loop Gazebo simulation, we use the *Iris* quadcopter with PX4 firmware providing the simulation tools. We emphasize how our method works and integrates well with both hexacopter and quadcopter systems.

The full state of the system, whether it is a hexacopter or quadcopter, is given by $x = [p_x, p_y, p_z, v_x, v_y, v_z, q_w, q_x, q_y, q_z, \omega_x, \omega_y, \omega_z]$, where $p_W = [p_x, p_y, p_z]$ is the position in the world frame, $v_W = [v_x, v_y, v_z]$ is the velocity in the world frame, $q_{WB} = [q_w, q_x, q_y, q_z]$ is the unit quaternion representing the body orientation, $\omega_B = [\omega_x, \omega_y, \omega_z]$ is the angular rate in the body frame, and the world and body frames follow respectively the traditional East-North-Up and Forward-Left-Up, shown in Figure 15. The state is estimated at a frequency of 100 Hz using the PX4 implementation of an Extended Kalman Filter that fuses the measurements from the onboard IMU and our Vicon motion capture system. Besides the flight controller that handles the state estimation and motor control, the main computational unit on the hexacopter is a *Beelink MINIS 12*. Its primary task is to compile our neural SDE models implemented in JAX, receive state estimates from the flight controller, solve the stochastic NMPC, and send back the motor commands and desired angular rate to the flight controller for low-level motor control. In software in the loop simulation, we could track desired trajectories via sending motor commands directly output from our MPC. However, due to the latency during hardware experiments, we instead send desired angular rates from the NMPC to the flight controller, which then uses a PI controller to track the desired angular rates.

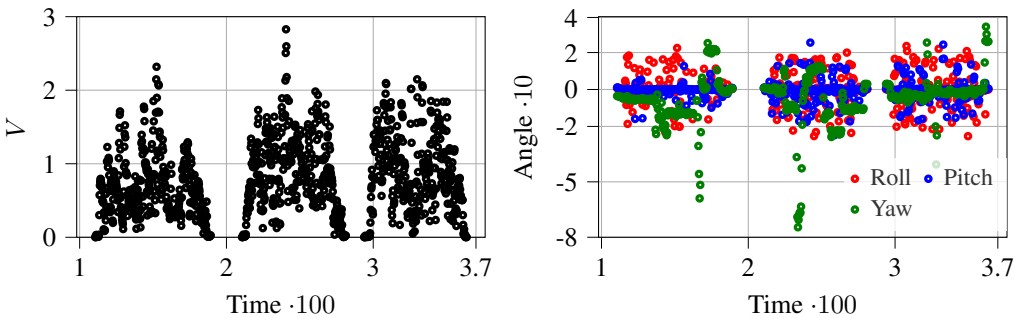

Figure 16: **Hexacopter system**: The velocity magnitude and Euler angles attained during data collection. The hexacopter mostly operates in the low speed and low Euler angles regime.

**Data Collection.** We seek a predictive model for the hexacopter and quadcopter dynamics that can be used to autonomously track aggressive trajectories. To this end, we collect 3 system trajectories by manually flying the hexacopter via a radio-based remote controller. For the quadcopter in Gazebo, we use a Joystick controller to collect *three* trajectories of the system. During data collection, we store the estimated state $x$ at a frequency of 100 Hz, as well as the desired motor input commands $u = [u_1, u_2, u_3, u_4, u_5, u_6]$ for the hexacopter and $u = [u_1, u_2, u_3, u_4]$ for the quadcopter. Figure 16 shows the velocity magnitude and Euler angles from the collected dataset in the case of the hexacopter hardware experiments. We obtain a total of 203 seconds worth of flight data, with the 3 trajectories being respectively 73, 66, and 64 seconds long. Besides, Figure 16 shows that 95% of the collected data corresponds to the hexacopter operating below the speed of 1.71m/s, absolute roll of 18°, absolute pitch of 13°, and absolute yaw of 24°. Instead, the maximum absolute speed, roll, pitch, and yaw attained are respectively 2.7 m/s, 23°, 23°, and 80°.

**Benchmark Models.** We use the proposed neural SDE framework to train a model of the hexacopter and quadcopter dynamics with the limited dataset described above. Our neural SDE model takes advantage of the general structure of 6-dof rigid body dynamics while having as unknown terms: The aerodynamics forces and moments, the motor command to thrust function, and (geometric) parameters of the system such as the mass and the inertia matrix. Specifically, our physics-informed neural SDE model is given by:

$$
\mathrm{d}\begin{bmatrix} p_W \\ v_W \\ q_{WB} \\ \omega_B \end{bmatrix} = \begin{bmatrix} v_W \\ \frac{1}{m_\theta}\big(q_{WB}\big(T_\theta(u) + f_\theta^{\mathrm{res}}(x^{\mathrm{feat}})\big)\bar{q}_{WB}\big) + g_W \\ \frac{1}{2}q_{WB}\omega_B \\ J_\theta^{-1}\big(M_\theta(u) + M_\theta^{\mathrm{res}}(x^{\mathrm{feat}})\big) - \omega_B \times J_\theta\omega_B \end{bmatrix} \mathrm{d}t + \sigma^{\mathrm{max}} \odot h_\phi(\eta_\psi(x^{\mathrm{feat}})) \star \mathrm{d}W,
$$

(4)

where $x^{\mathrm{feat}} = [v_W, \omega_B]$, $\times$ denotes the cross product, $\bar{q}_{WB}$ is the conjugate of $q_{WB}$, the product $qv$ between a quaternion $q$ and a vector $v$ is define as the quaternion product between $q$ and the 4-D vector $[0; v]$, the vector $\sigma^{\mathrm{max}} = [1, 1, 1, 10, 10, 10, 1, 1, 1, 50, 50, 50] \cdot 10^{-3}$ is the maximum diffusion term, the variables $m_\theta$ and $J_\theta = \mathrm{diag}(J_\theta^{\mathrm{x}}, J_\theta^{\mathrm{y}}, J_\theta^{\mathrm{z}})$ represent the system mass and inertia matrix, the neural network functions $f_\theta^{\mathrm{res}}$ and $M_\theta^{\mathrm{res}}$ represent the residual forces and moments due to unmodelled and higher order aerodynamic effects, the parametrized functions $T_\theta$ and $M_\theta$ provide estimate of the motor command to thrust and moment values, and $g_W = [0, 0, -9.81]^\top$ is the gravity vector. Specifically, we parametrize $f_\theta^{\mathrm{res}}$, $M_\theta^{\mathrm{res}}$ as feedforward neural networks with $\tanh$ activation functions and 2 hidden layers of size 8 and 16, respectively. The motor thrust forces and moments are learned via $[T_\theta^\top, M_\theta^\top]^\top = [0, 0, T_\theta^{\mathrm{z}}, M_\theta^{\mathrm{x}}, M_\theta^{\mathrm{y}}, M_\theta^{\mathrm{z}}]^\top = A_\theta^{\mathrm{mix}}[T_\theta^{\mathrm{mot}}(u_1), \ldots, T_\theta^{\mathrm{mot}}(u_6)]^\top$, where $A_\theta^{\mathrm{mix}}$ is a $6 \times 6$ matrix of learnable parameters constrained by the geometry of the hexacopter, and $T_\theta^{\mathrm{mot}}$ is a parametrized function that maps the motor commands to the thrust forces. We use polynomial functions for $T_\theta^{\mathrm{mot}}$ and empirically found that a degree of 1 as $T_\theta^{\mathrm{mot}}(z) = \alpha_\theta z + \beta_\theta$ works particularly well for control purpose compared to higher order polynomials. We emphasize the diffusion terms on $p_W$ and $q_{WB}$ are low as their dynamics are known, and the noise in the estimation will come from integrating the noisy velocity and angular rate components. For training the neural SDE model, we use the derivative-free Milstein method as the SDESolve algorithm with a step size of 0.05 and a horizon $n_{\mathrm{r}} = 20$. Further, we use $\lambda_{\mathrm{sc}} = 1$ for encouraging large strong convexity coefficients and a vector of ball radius $r = 0.1$ to enforce the strong convexity property locally. The quadcopter modeling is the same as for the hexacopter, with the thrust forces and moments being a function of the 4 control inputs instead of 6 for the hexacopter.

To illustrate the prediction accuracy of our model, we compare it with a system identification-based approach that uses the same formulation as our SDE model (4) but without the diffusion term and the residual neural network terms $f_\theta^{\mathrm{res}}$ and $M_\theta^{\mathrm{res}}$. Precisely, with system identification, we seek to identify all the parameters $m_\theta$, $J_\theta$, $A_\theta^{\mathrm{mix}}$, $\alpha_\theta$, and $\beta_\theta$ by estimating $\dot{x}$ from the dataset using finite difference and then solving a least square problem to fit the system's differential equation to the data. This is a simple approach for modeling multirotors that has been widely used in the literature. Additionally, as another baseline, we train a neural ODE model with the same architecture as the neural SDE model but without the diffusion term.

**Nonlinear Model Predictive Control.** We seek to use our learned SDE for autonomous control of the hexacopter. To this end, we employ a receding horizon model predictive control approach.

Such an approach uses the learned SDE model to predict future system trajectories over a fixed time horizon, and then optimize the control inputs to minimize a cost function that penalizes the control effort and deviation from a reference trajectory. For numerical optimization, we discretize the state and control inputs into $n_{\mathrm{r}} = 20$ equal time intervals over the horizon $H = 1$ second, yielding a constrained optimization problem of the following form at each state measurement $x_{\mathrm{init}} = x_t$:

$$\underset{u_1,\ldots,u_{n_{\mathrm{r}}}}{\text{minimize}} \quad \mathbb{E}\Big[\sum_{k=1}^{n_{\mathrm{r}}}(x_k - x_k^{\mathrm{ref}})Q(x_k - x_k^{\mathrm{ref}}) + u_k R u_k\Big] \tag{5}$$

$$\text{subject to} \quad \{x_1^p,\ldots,x_{n_{\mathrm{r}}}^p\}_{p=1}^{n_{\mathrm{p}}} = \text{SDESolve}(x_0, u;~(4)), \tag{6}$$

$$x_0 = x_{\mathrm{init}},~u_1,\ldots,u_{n_{\mathrm{r}}} \in [0,1], \tag{7}$$

where $x_k^{\mathrm{ref}}$ is the reference state at time $t_k = t + kH/n_{\mathrm{r}}$, we use $q - q^{\mathrm{ref}}$ to denote the term $(q(q^{\mathrm{ref}})^{-1})_{xyz}$, the positive definite matrices $Q$ and $R$ penalize the deviation from the reference trajectory and the control effort, respectively, and $n_{\mathrm{p}}$ is the number of particles used for the SDE solver. In all our experiments, we used $n_{\mathrm{p}} = 1$, $Q = \text{diag}(100, 100, 200, 5, 5, 10, 1, 1, 100, 1, 1, 1)$, and $R = \text{diag}(1, 1, 1, 1, 1, 1)$. Besides, we solve the above optimization problem using an adaptive learning rate, Nesterov acceleration-based projected gradient descent; all implemented in JAX.

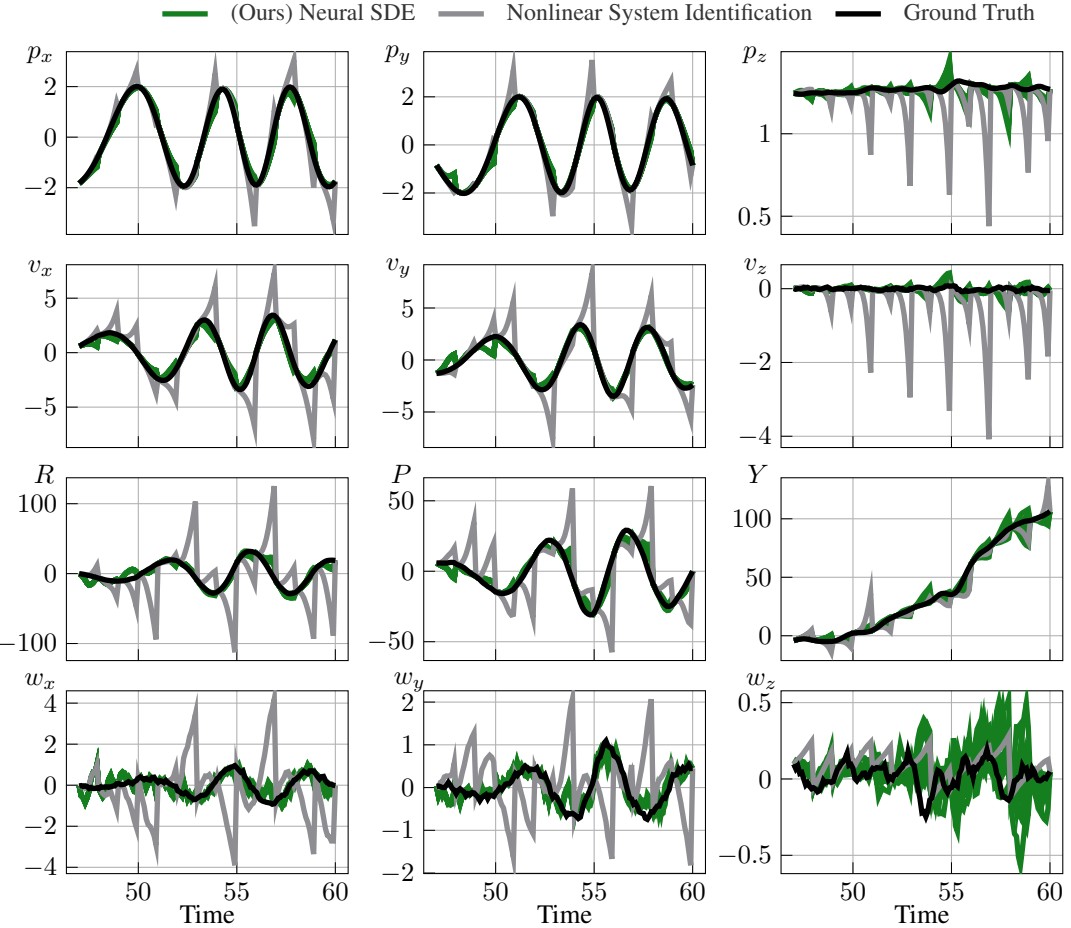

Figure 17: **Hexacopter system**: Time evolution of the hexacopter state predicted by the learned SDE model and the vanilla system identification-based model. The quantities $R, P, Y$ denote the roll, pitch, and yaw angles, respectively, with units in degrees.

**Hexacopter modeling: Neural SDE improves prediction accuracy over system identification while providing uncertainty estimates.** Figure 17 is a complete version of Figure 7, where the prediction of the remaining states of the system is also provided. The neural SDE model provides

uncertainty estimates while also providing better long-horizon prediction capabilities compared to the system identification-based model.

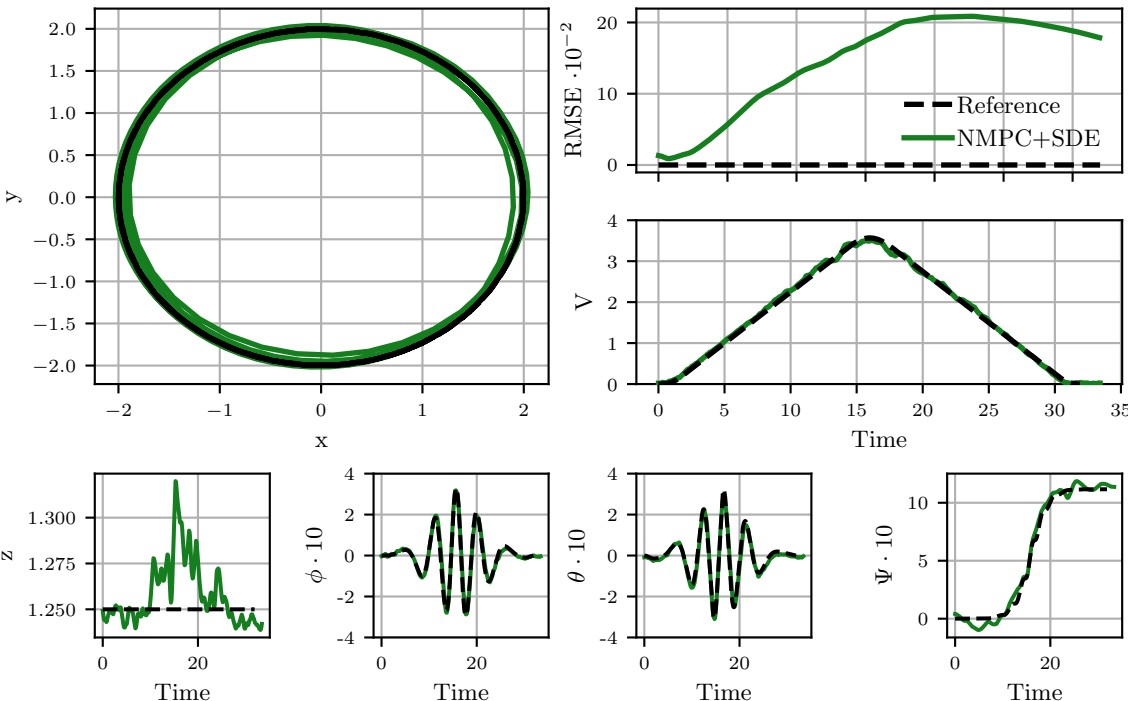

Figure 18: **Hexacopter system**: Tracking performance achieved on the circle trajectory. The MPC based on our learned SDE achieves an RMSE of 20 cm for a 35 seconds trajectory while the hexacopter reaches speed up to 3.6 m/s, roll and pitch angles up to 32°, and yaw angle up to 120°.

**Hexacopter control: Neural SDE yields high-performance MPC algorithms even when operating far beyond what it has been trained on.**   Figure 18 and Figure 19 show the results of our experiments on a circle trajectory and lemniscate trajectory, respectively. These reference trajectories are obtained by minimum snap trajectory generation [65] without any prior knowledge of the hexacopter dynamics. For both trajectories, we show the time evolution of the velocity, roll, pitch, yaw, and tracking accuracy during autonomous control. These plots demonstrate the ability of our learned SDE to generalize far beyond what it has been trained on. In fact, we can observe that the hexacopter must reach velocity up to 3.6 m/s, roll and pitch angles up to 32°, and yaw angle up to 120°, in order to track the reference trajectories. We emphasize how these values are outside of the training data regime, as shown and detailed in the data collection section. Despite this, the learned SDE is able to generalize to these extreme conditions and achieves a high tracking accuracy of 20 cm and 15 cm for the circle and lemniscate trajectories, respectively. We also note that the tracking accuracy is not uniform across the trajectory, and it is lower when the hexacopter is moving faster. This is expected as the hexacopter is more difficult to control when it is moving faster. Besides, the performance of our control approach is further displayed on the plot showing the evolution of the altitude, where an altitude error of less than 5 cm is achieved consistently.

**Quadcopter Gazebo simulation: Neural SDEs improve model accuracy and model predictive control performance compared to neural ODEs and system identification baselines.**   Figure 20 shows the predictive capabilities of the neural SDE and baseline models on the trajectories collected from the Gazebo simulation of the Quadcopter. The plot shows a better approximation of the system identification-based models due to a less noisy dataset than the real-world hexacopter collected dataset. We observe that the neural ODE improves prediction accuracy over the system identification-based techniques but has slightly lower accuracy than the neural SDE model, specifically for extended integration steps.

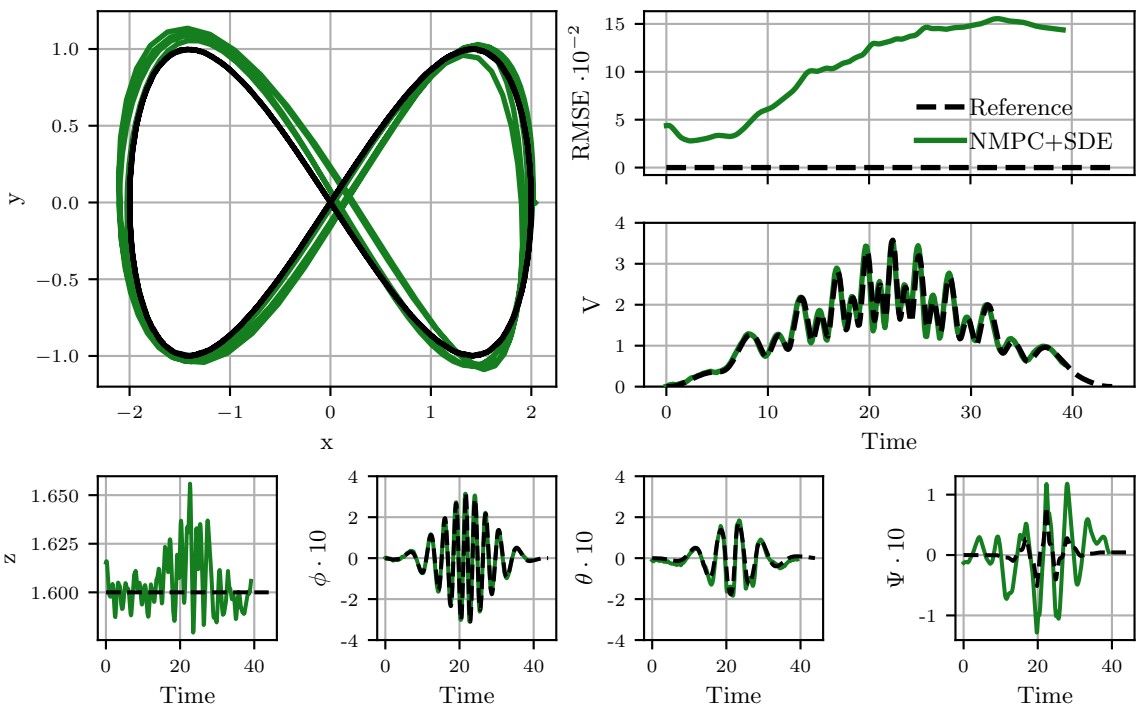

Figure 19: **Hexacopter system**: Tracking performance achieved on the lemniscate trajectory. The MPC based on our learned SDE achieves an RMSE of $15$ cm for a $40$ seconds trajectory while the hexacopter reaches speed up to $3.4$ m/s, roll angle up to $32°$, pitch up to $19°$, and yaw up to $13°$.

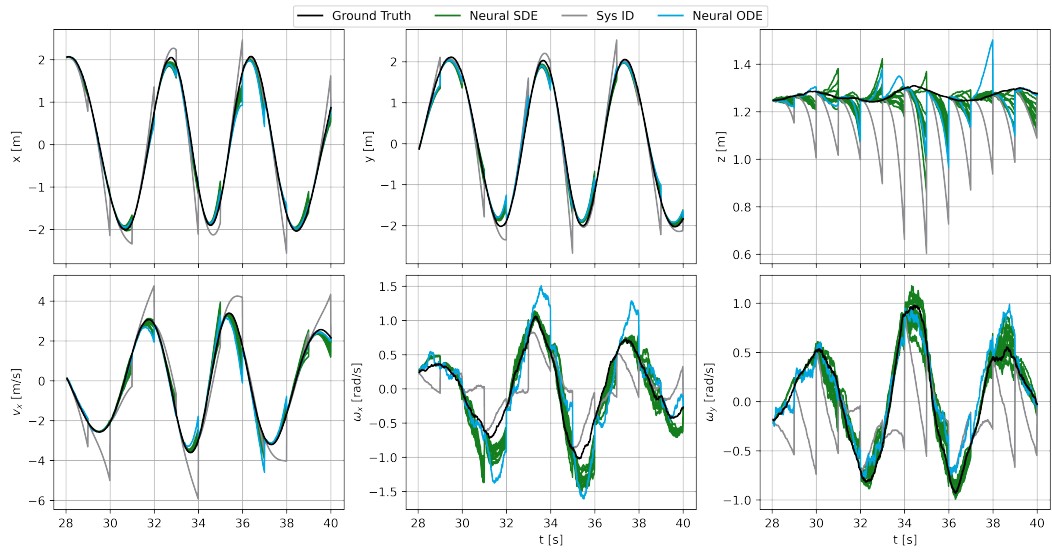

Figure 20: **Quadrotor Gazebo simulation**: Comparison of the prediction accuracy of neural SDE, ODE, and system identification-based model.

Moreover, Figure 21 and Figure 22 show the results using the different learned models for predictive control in Gazebo simulation. We observe that the neural SDE-based controller more accurately tracks the specified trajectories than the SysID-based controller. The gap between their tracking accuracies appears to be largest when the trajectories require the highest velocities and the largest

angular rates. This observation indicates that the neural SDE has relatively high predictive accuracy, particularly in regions of the state space that are beyond the training dataset. Besides, we observe a slight improvement in control performance between the neural SDE and neural ODE model.

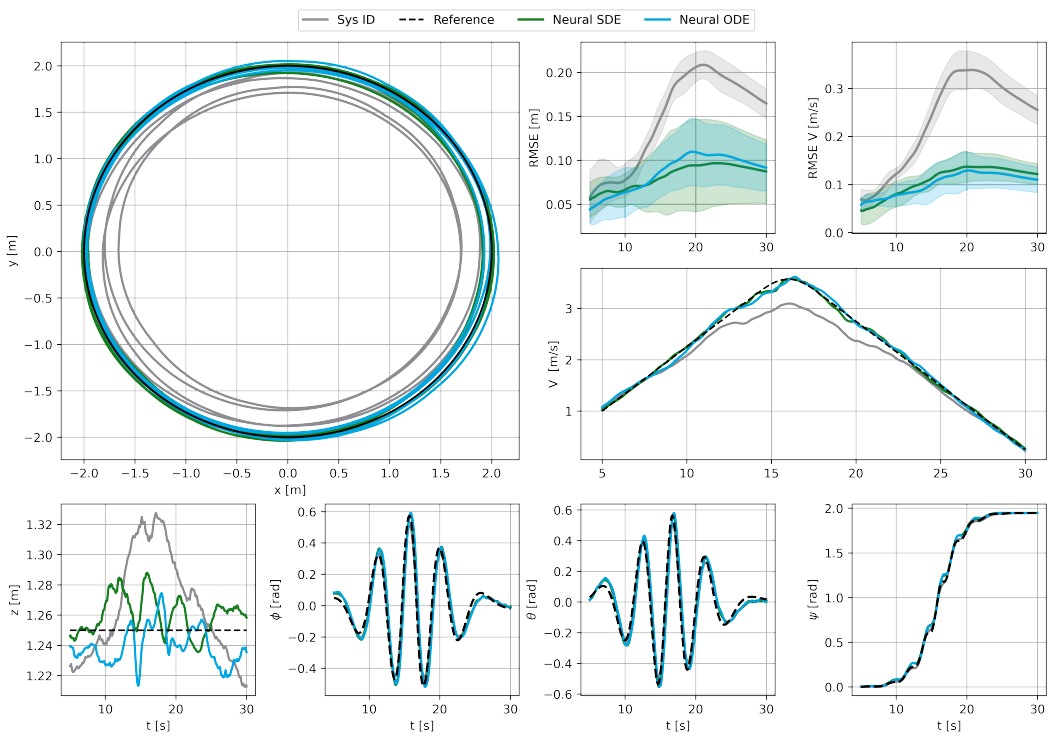

Figure 21: **Quadrotor Gazebo simulation**: Tracking performance on the Circle trajectory.

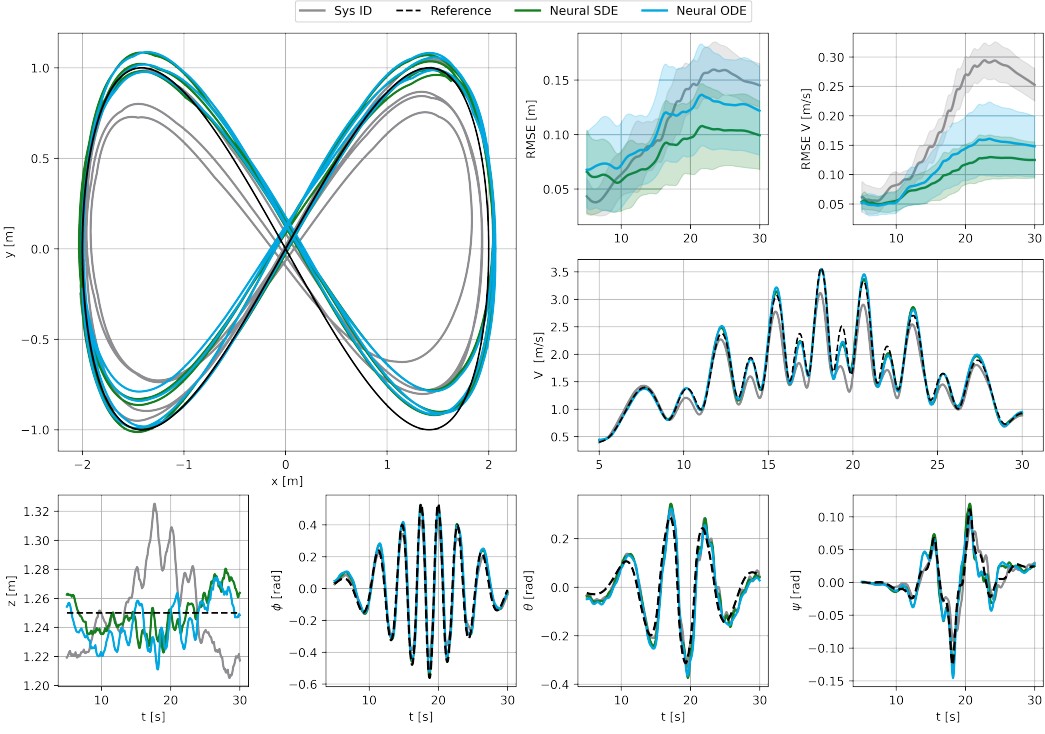

Figure 22: **Quadrotor Gazebo simulation**: Tracking performance on the lemniscate trajectory.

