# OpenReview forum: "How to Learn and Generalize From Three Minutes of Data: Physics-Constrained and Uncertainty-Aware Neural Stochastic Differential Equations"
_robot-learning.org/CoRL/2023/Conference — CoRL 2023 Oral_

### Official Review · Reviewer_DHcC · 2023-07-19

**Confidence:** 5
**Originality:** Very Good
**Technical Quality:** Excellent
**Clarity Of Presentation:** Very Good
**Impact:** 4

**Recommendation:**

Strong Accept: I recommend accepting the paper and will argue for my recommendation even if other reviewers hold a different opinion.

**Review:**

The paper presents a novel approach to model-based learning and control using a neural stochastic differential equation. The key insight of the work is that effective modeling of low-data regimes is key to have conservative behavior when unsure and confident, aggressive maneuvers high-data regimes.

A strength of the work is the empirical evidence tied to the theoretical justification for the proposed work. The sufficient ablation studies compare with existing methods for model learning and demonstrate clear improvement.

One weakness that the paper addresses is the single-use case with the drone. Though it is possible to extend this relatively quickly. In addition, the paper does assume full observability which is highlighted in the limitations.

Another, perhaps more critical weakness of the work is the choice of metric for the uncertainty. The paper assumes a Euclidean metric is sufficient to model distance in the model variance. However, Euler angles are not Euclidean and do not exhibit the properties of a vector space. This can lead to issues with large angles and misrepresented uncertainty values. It is critical that the paper address this assumption in the limitations as this could change the problem structure all together.

**Quality Of The Limitations Section:**

Additional details required

**Questions For Rebuttal:**

Is the proposed approach amenable to other metric choices?

At what limit does the approach fail to capture the true dynamics (is this a function of data or metric choices)?

**Robotics Focus:**

Sufficient demonstration on hardware

**Summary Of Paper:**

The paper presents a method for learning dynamics models using physics constrained neural stochastic differential equations. The approach demonstrates that one can learn meaningful uncertainty models in low-data regimes while accurately predicting drone dynamics with just three minutes of data and physics priors. The physical robot results are convincing and demonstrate the effectiveness of the proposed approach.

**Summary Of Recommendation:**

The work demonstrates a novel approach to model-based control using a neural stochastic differential equation. The results are impressive and demonstrate clear applicability on a physical system. The comments mentioned earlier should be addressed to highlight the limitations of the proposed work, but the approach appears novel and exciting. Based on the rebuttal and responses, an accept is recommended.

---

### Official Review · Reviewer_BxRN · 2023-07-20

**Confidence:** 4
**Originality:** Very Good
**Technical Quality:** Very Good
**Clarity Of Presentation:** Very Good
**Impact:** 4

**Recommendation:**

Weak Accept: I recommend accepting the paper, but will not argue for my recommendation if the majority of other reviewers have a different opinion.

**Review:**

The experimental results are undeniably impressive and the paper is well-written with clearly described insights derived from each of the experiments. I am pleased to recommend its acceptance; please find below a list of comments/questions that if addressed may help readers better appreciate the impact of this work.
- *SDE training loss:* The loss (2) sums up the log-likelihoods of the data with respect to each fixed-variance Gaussian centered around integrated SDE paths. Though it's noted that this is a "common choice in the literature [33, 30]" it seems that this treatment of likelihood may misrepresent multimodal data distributions (i.e., in such a case I believe the SDE would learn to predict the mean -- indeed, I'm not sure how (2) encourages the model to capture any aleatoric uncertainty), though notably none of the systems considered in this work exhibit multimodal uncertainty. That is, I had initially expected that (2) would essentially be using a set of particles sampled from the SDE dynamics (i.e., $\text{SDESolve}$) to produce a kernel density estimate of the SDE's trajectory distribution pdf which would then be used to evaluate the likelihood of the observed data (when implemented this would be not unlike (2), just with a logsumexp). In any case some additional explanation/justification would be beneficial for readers to understand why this loss formulation is chosen.
- *Drift term architecture:*
  - The interpretation of $\eta_\psi$ and $h_\phi$ is not completely clear to me -- if $\eta_\psi$ is intended to encode distance to the training dataset why does it need to saturate at 1 (especially when $h_\phi$, which also saturates at 1, immediately applies a $\text{sigmoid}^{-1}$ to the output of $\eta_\psi$)?
  - Is there an advantage to representing aleatoric and epistemic uncertainty jointly? That is, one might imagine decomposing the total uncertainty into a sum of two terms: (i) a non-negative aleatoric uncertainty term and (ii) a distance-based epistemic uncertainty term that is encouraged to take zero values on points in the training dataset (compared to zero-gradient, as for $\eta_\psi$, which requires the strong convexity loss to avoid saddle points).
- *Hexacopter MPC comparison:*
  - In SysID what degree polynomial basis is used to fit the nonlinear unknown functions (mentioned on line 322)? I believe that with appropriate constants for $m$ and $J$, linear gains defining $T_\theta$ and $M_\theta$, and assuming $f^{res} = M^{res} = 0$ roboticists have historically been able to implement multicopter tracking MPC controllers. Notably only open loop predictions are compared in Figure 7 (where perhaps the erratic SysID predictions may be attributed to Runge's phenomenon with high-order polynomial fitting). Is 3 minutes also enough for a traditional SysID + MPC approach to track trajectories (though perhaps with greater error)?
  - Though neural SDEs are topical today, it is not clear that this method provides benefit over other state-of-the-art approaches, e.g., [R1] which uses Gaussian processes that incorporate a priori physics knowledge and provide estimated uncertainty, similar to the motivations of this work.
- *Stochastic NMPC implementation:* Though more detail is provided in the Appendix, I believe that some concept of how the stochastic NMPC is implemented should be stated in the main text (i.e., that a control sequence is chosen to minimize expected cost over SDE predicted trajectories, estimated empirically using samples without explicitly taking any uncertainty estimates (e.g., diffusion term magnitude) into account).

[R1] G. Torrente, E. Kaufmann, P. Fohn, and D. Scaramuzza, "Data-Driven MPC for Quadrotors," *IEEE Robotics and Automation Letters*, 2021. Available at https://arxiv.org/abs/2102.05773.pdf.

**Quality Of The Limitations Section:**

Limitations are addressed clearly

**Questions For Rebuttal:**

From the comments/questions above, please prioritize:
- justification of the loss formulation Eq. (2), and
- describe the performance (at least in simulation) of the SysID baseline when deployed in closed-loop MPC.

**Robotics Focus:**

Sufficient demonstration on hardware

**Summary Of Paper:**

In this paper the authors propose a structured dynamics learning methodology that fits neural stochastic differential equations (SDEs) to observed trajectory data. The SDE consists of a drift term and diffusion term capturing the deterministic and stochastic components of the dynamics respectively; the authors propose that the design of the drift term can be physics-based with unmodeled/error terms parameterized as neural networks, and that the diffusion term may capture both the aleatoric and epistemic uncertainty (the latter estimated using distance to training data). The resulting learned dynamics may be employed in model-based RL or model predictive control algorithms to provide safe, uncertainty-aware policies even in the absence of full data coverage. Experiments on simple simulated systems as well as in hardware on a hexacopter system tracking aggressive trajectories demonstrate the improved performance over baselines and data efficiency of the approach.

**Summary Of Recommendation:**

This paper achieves its stated goal of demonstrating the capability of neural SDEs to capture model uncertainty while incorporating strong inductive bias; these aims are of broad relevance to the robot learning research community and therefore I recommend this paper's acceptance at CORL 2023. The experiments provide convincing evidence for this method's potential, though I believe that from a practitioner's perspective additional evaluation may be needed to show that this approach should be selected over other state-of-the-art model-based methods.

---

### Official Review · Reviewer_5AP6 · 2023-07-21

**Confidence:** 4
**Originality:** Fair
**Technical Quality:** Good
**Clarity Of Presentation:** Very Good
**Impact:** 3

**Recommendation:**

Weak Accept: I recommend accepting the paper, but will not argue for my recommendation if the majority of other reviewers have a different opinion.

**Review:**

Overall, I find the main idea of the paper to be clear -- SDEs have been shown to be capable of modeling epistemic uncertainty (Kong et al, ICML 2020), and so it is natural to attempt to learn unknown dynamics, with additional requirements of uncertainty modeling. Neural ODEs (NODEs) and SDEs also allow for easy incorporation of prior knowledge, by having a mixture of trainable apriori specified terms to represent terms in the ODE or SDE. My questions and concerns are as follows:

- The empirical evaluation is critically underwhelming. It is unclear whether the benefits of the uncertainty estimates provided by the NSDE are of any benefit, over a NODE with its dynamics parameterised in the same way as the drift term of the SDE. NSDEs are much more unstable to train and slower to roll out than NODEs -- *is the heavy machinery worth it, wrt to NODes?* Would love to see examples where the NODE fails and some discussion of this.

- Also critically lacking is the comparison against Gaussian Process-based approaches. The uncertainty estimates for NSDEs were visually compared against ensemble-based approaches to extract uncertainty. Gaussian processes have been well studied and are the gold standard for epistemic uncertainty quantification. Additionally, GPs have been used within an MPC setting for race cars (Hewing, Kabzan, and Zeilinger) and drones (Liu, Shi, Chung, Anandkumar, Yue, L4DC 2020), to learn unknown parameters (with additional uncertainty estimates available) within a controller. It would be good to explore the trade-offs between GPs and NSDEs -- why NSDEs over GPs?

Overall the paper is fairly well-written, and the method appears sounds -- I am happy to increase my score after these concerns have been made.

**Quality Of The Limitations Section:**

Additional details required

**Questions For Rebuttal:**

- Justification of NSDEs over NODEs, and some clarity over the computation (and also training stability) trade-offs between using an NSDE over a NODE. Please list out inference and training times wrt to NSDEs and NODEs, and ideally discuss an example where NSDEs show marked improvement over NODEs. NODEs have not been compared on the use-case problems, so it is unclear to the reader the practical benefits of NSDE over NODE.

- Comparisons and/or discussions against previous Gaussian Process-based approaches to similar problems.

**Robotics Focus:**

Sufficient demonstration on hardware

**Summary Of Paper:**

This paper proposes to use Neural Stochastic Differential Equations  (NSDEs), i.e. a differentiable SDE integrator, to learn dynamics models from data. A main motivation for this is the ability of NSDEs to be distance-aware. Specifically, the model is uncertain about states that are deemed far from the data distribution. The drift and diffusion terms of the NSDE can parameterised as trainable models.

**Summary Of Recommendation:**

Please see the review above.

---

### Official Review · Reviewer_S722 · 2023-07-23

**Confidence:** 3
**Originality:** Good
**Technical Quality:** Good
**Clarity Of Presentation:** Good
**Impact:** 3

**Recommendation:**

Weak Accept: I recommend accepting the paper, but will not argue for my recommendation if the majority of other reviewers have a different opinion.

**Review:**

## clarity

The paper is mostly clear, up to a few things that I think need to be elaborated (see the questions). There is a good amount of experiments that support the claim that the paper is making, that this Neural SDE formulation is good for accounting for uncertainty and for being used as a model.

## strengths

The work is novel to the extent of my knowledge, providing an inductive bias to the drift term in order to improve dynamics modelling is well-motivated, also the theoretical framework for dealing with the problem of uncertainty estimation is correct.

Experimental evaluation validates the claims of the paper.

There are results on real hardware.

## weaknesses

clarity can be improved regarding the computation of the diffusion term and the experiments.

the diffusion coefficient seems overly engineered

some claims in the paper might be wrong (see comments)

**Quality Of The Limitations Section:**

Limitations are addressed clearly

**Questions For Rebuttal:**

you should cite work from [1],  Indeed, your reference to PETS considers only sampling, but doesn't separate uncertainties and doesn't take them into account, whereas Risk Averse Trajectory Optimization that builds on top of it does.

eq. 2 - the likelihood of the trajectory is indeed not Gaussian, this is not the correct likelihood given the Neural SDE, therefore it's not an NLL loss given oyour probabilistic model. Likelihood computation can be done by probability flow ODE + instant change of variables. Or am I missing something here? This is very critical.

line 138 - what do you mean by elementwise maximum values for the $\Sigma_{max}$? Is this just the maximum over the concatenated vector [x,u]?

paragraph line 139 - this indeed enforces local minima, but it doesn't enforce that no additional uncertainty is introduced when you are on-data?

figure 4 - nice figure

figure 5 - why is there no error band for the prediction error here? Also, how is the prediction error computed for the Neural SDE? Since it is a sampling method, do you make an MC estimate of the expected error? Over how many samples? The evaluation metrics are not very clear to me.

figure 6, PPO experiment - why didn't you report results for the neural ODE case? Except being curious about them, I believe that you need them for completeness.

[1] Vlastelica, Marin, Sebastian Blaes, Cristina Pinneri, and Georg Martius. "Risk-averse zero-order trajectory optimization." In 5th Annual Conference on Robot Learning. 2021.

**Robotics Focus:**

Sufficient demonstration on hardware

**Summary Of Paper:**

The paper proposes to use neural SDEs to model robot dynamics that consist of an informed drift term (governing equations, rigid body dynamics and learned dynamics parameters) and  diffusion term that captures the uncertainty of the dynamics. The main idea behind the diffusion term is that uncertainty is greater when prediction come farther from the data. The term is learned by enforcing local minima in regions of the data and strong convexity of the within a ball around the data points.

**Summary Of Recommendation:**

The main thing that is bothering me (why it is a weak reject) is the way that the predictive trajectory distribution is treated (see the rebuttal questions) + secondary things.

---

### Decision · Program_Chairs · 2023-08-30

**Decision:**

Accept (Oral)

**Comment:**

The paper presents a technique to learn the system dynamics and associated uncertainty using neural stochastic differential equations where one term models the drift and the other captures the diffusion. This makes the model able to capture the epistemic uncertainty when inferring points far from the data distribution. Experiments were performed on simple simulated systems as well as on a real hexacopter.

Reviewers were positive about the novelty of the work but were concerned about the experimental section and lack of comparisons to equivalent formulations such as GP-based dynamical systems. Authors have mostly addressed these concerns and have elaborated on the limitations of the work in the revised version.